# CAUSAL REASONING FAVORS ENCODERS: ON THE LIMITS OF DECODER-ONLY MODELS

## ABSTRACT

In-context learning (ICL) underpins recent advances in large language models (LLMs), although its role and performance in causal reasoning remains unclear. Causal reasoning demands multi-hop composition and strict conjunctive control, and reliance on spurious lexical relations of the input could provide misleading results. We hypothesize that, due to their ability to project the input into a latent space, encoder- and encoder–decoder architectures are better suited for said multi-hop conjunctive reasoning versus decoder-only models. To do this, we compare fine-tuned versions of all the aforementioned architectures with zero- and few-shot ICL in both natural-language and non-natural language scenarios. We find that ICL alone is insufficient for reliable causal reasoning, often overfocusing on irrelevant input features. In particular, decoder-only models are noticeably brittle to distributional shifts, while fine-tuned encoder and encoder–decoder models can generalize more robustly across our tests, including the non-natural language split. Both architectures are only matched or surpassed by decoder-only architectures at large scales. We conclude by noting that for cost-effective, short-horizon robust causal reasoning, encoder or encoder-decoder architectures with targeted fine-tuning are preferable.

## 1 INTRODUCTION

Causal reasoning is a foundational capability for modern AI systems. It underpins scientific inference, reliable decision making under interventions, and safety-critical applications where spurious correlations could lead to inaccurate predictions, and thus brittle and unsafe behavior. A core component of causal reasoning is the ability to carry out structured, rule-based deduction: to combine multiple premises, enforce boundary conditions, and derive correct conclusions in a way that remains stable under representational changes. This deterministic, implication-driven form of reasoning, often formalized as logical deduction in first-order logic (FOL) is a crucial building block of many causal systems. Large language models (LLMs), when prompted with examples, often appear capable of such structured inference through in-context learning (ICL), yet it is unclear how reliably they implement it. Logical reasoning imposes two stringent requirements: multi-hop composition and strict conjunctive control. The former is the ability to chain several elementary implications or constraints to derive a conclusion across multiple intermediate steps. The latter requires the issuance of a positive decision only when all relevant premises, guards, and boundary conditions are simultaneously satisfied; and rejected otherwise.

It is known that LLMs exhibit limitations in logical reasoning, including difficulty in being logically consistent (Calanzone et al., 2024) memorization Xie et al. (2024) and reliance on spurious lexical features (Zhang et al., 2022; Sanyal et al., 2022).These shortcomings are particularly visible in natural language settings (Parmar et al., 2024).For example, Han et al. (2024a) demonstrated that LLMs struggle with FOL reasoning. Although LLMs can often perform step-level reasoning, they struggle with proof planning and choosing the correct next step when multiple options are available, and their generalization remains uneven- robust on compositional proofs yet unreliable on unseen deduction rules (Saparov & He, 2022; Saparov et al., 2023).These behavioral failures naturally raise the question of whether architectural choices fundamentally constrain LLMs' logical reasoning abilities. Prior analyses (Tian et al., 2021; Han et al., 2024a) have compared a wide range of architectures on FOL reasoning tasks and found significant differences across model families. However, there has

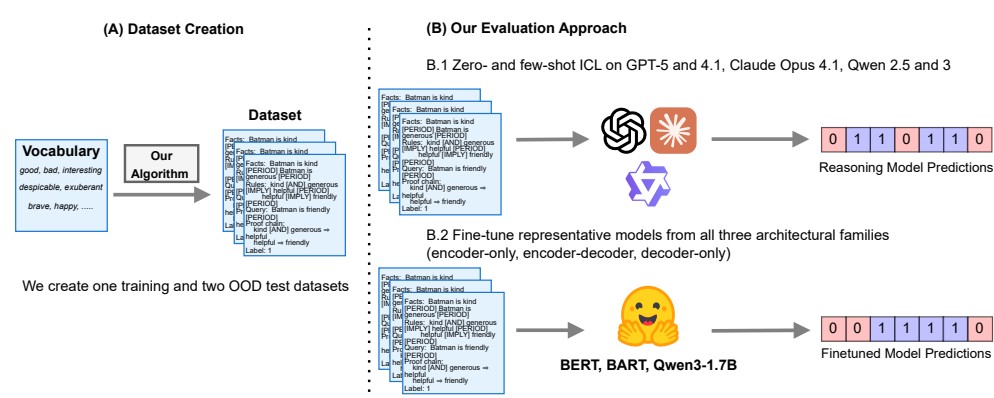

Figure 1: Overview of our approach. *Left, (A)*: Dataset creation. From a fixed set of propositions, we generate one training set, and two out-of-distribution (OOD) test sets. *Right, (B)*: For our evaluation we conduct zero- and few-shot inference with decoder-only reasoning and non-reasoning models. We also fine-tune models from all three architectural families (encoder-only, encoder–decoder, and decoder-only); namely, variants of BERT and BART, and Qwen3-1.7B (Yang et al., 2025)

not been a systematic comparison between decoder-only models[1] and encoder-enabled architectures such as BERT (Devlin et al., 2019) and BART (Lewis et al., 2019).

In this work we compare encoder and encoder–decoder architectures with decoder-only architectures. Encoder-based architectures have the innate ability to project the full input into a latent space; while decoder-only architectures perform inference recursively, in a token-by-token fashion. Our hypothesis is that an encoder's projective ability will allow them to perform more reliable causal composition, especially under distributional shifts, when compared to decoders. To test this, we evaluate a series of encoder, decoder, and encoder-decoder architectures under various out-of-distribution (OOD) scenarios, with synthetic data especially designed to evaluate their robustness to distributional shifts.[2] We test these distributional shifts by (1) progressively deeper reasoning chains in a subset of first-order logic (FOL), and (2) the same dataset with randomized characters to ablate out lexical relations (e.g., "Batman is kind" in the first would be "Batman is a#d}" in the second).

We find that, under our setup, encoder-only models are able to generalize better than decoders. This means that they are able to have better accuracies at deeper reasoning chains, as well as learn to focus on the underlying logical structures of the data. This latter part holds both for non-finetuned (i.e., zero- and few-shot) and finetuned decoder-only models. Moreover, this performance gap becomes wider as the reasoning chains in both scenarios become deeper. The only outlier to our findings is a recent, state-of-the-art (SOTA) model, GPT-5 (OpenAI, 2025b), which attained near-perfect accuracy in all our tests. However, this incurred substantial latency and–presumably–higher compute cost. We thus conclude that **ICL alone is not an effective mechanism for causal reasoning**. Given the gaps in accuracy and compute cost, we argue that robust, cost-effective performance may be achieved with conventional encoder or encoder–decoder architectures and minimal fine-tuning.

## 2 RELATED WORKS

The study of logical reasoning capabilities in large language models (LLMs) has attracted significant attention in recent years. Several surveys provide broad entry points into this emerging area. See Liu et al. (2025) and Luo et al. (2023) for a review on analyzing and enhancing reasoning capabilities in LLMs. More recently, Roy et al. (2025) provided the first unified prompt and data-driven framework for full causal-graph discovery with LLMs, revealing that while in-context reasoning can approximate causal structure learning under favorable metadata conditions, current decoder-only ar-

---

[1]When the architecture is undocumented we assume the model discussed is a decoder-only LLM.

[2]All code and artifacts are available at https://anonymous.4open.science/r/causality_grammar-DB41/README.md

chitectures still struggle with strict compositional and conjunctive causal dependencies. Liu et al. (2025) offer an overview of current challenges and future directions in causal reasoning of LLMs.

**Architecture** Early investigations have explored causal reasoning in encoder-only and encoder–decoder architectures. For example, Pirozelli et al. (2024) study whether encoder-based models can handle logical reasoning tasks such as propositional and FOL, including validity checking and theorem proving. Dziri et al. (2023) explore how transformers solve compositional tasks that involve multi-step reasoning. Zheng et al. (2025) analyze the first order logical entailment abilities of both encoder-based and decoder only transformer models and find that they have comparable performance.

**Benchmarks** Prior work has introduced a range of logical reasoning benchmarks to assess the deductive abilities of LLMs. JustLogic (Chen et al., 2025) provides a large-scale synthetic benchmark designed to isolate deductive reasoning by removing prior knowledge dependencies and enabling fine-grained analysis across reasoning depth and argument structure, revealing large performance gaps between models and human ceilings. Cognitive science–inspired evaluations (Seals & Shalin, 2024) similarly show that current LLMs struggle with classical deductive reasoning problems when presented in their original form. Other efforts focus on improving evaluation diversity: DivLogicEval (Chung et al., 2025) highlights distributional biases in prior datasets and introduces linguistically diverse evaluation settings with metrics that reduce randomness and bias. In contrast to synthetic-only settings, FOLIO (Han et al., 2024a) and P-FOLIO (Han et al., 2024b) provide human-authored stories paired with first-order logic annotations and step-by-step proofs, enabling analysis of multi-step inference capabilities. Synthetic proof-oriented datasets like PrOntoQA (Saparov & He, 2022) also support structured formal evaluation through symbolic parsing of reasoning chains. Finally, LogicBench (Parmar et al., 2024) expands the scope of assessment by covering a wider range of propositional, first-order, and non-monotonic reasoning patterns, moving beyond the few inference rules traditionally tested. Together, these benchmarks reveal persistent weaknesses in LLM deductive reasoning despite growing capabilities.

**Improving logical reasoning in LLMs** Beyond evaluation, several methods have been proposed to enhance logical reasoning in LLMs. Logic-LM (Pan et al., 2023) augments language models with symbolic solvers to better navigate formal reasoning tasks by explicitly enforcing logical constraints. LogicAsker (Wan et al., 2024) takes a complementary skill-based approach, decomposing reasoning into atomic propositional and predicate logic abilities, and using these structured skills to diagnose and improve model performance. Data synthesis techniques have also proven effective: LogicPro (Jiang et al., 2025) creates a large corpus of challenging logic problems with verified reasoning chains, enabling substantial performance gains across multiple benchmarks such as LogicBench, GSM8K, and AR-LSAT. Meanwhile, DREAM (Cao et al., 2025) targets proof-based reasoning, identifying failures in multi-step first-order logic inference due to limited strategy exploration and error propagation. To address this, it introduces diversified proof generation and feedback-driven error correction, yielding notable improvements on complex theorem proving tasks. Collectively, these approaches demonstrate promising pathways to strengthen deductive reasoning in LLMs through external symbolic guidance, structured skill decomposition, data-driven training, and improved reasoning strategy search.

## 3 BACKGROUND

### 3.1 LOGICAL REASONING AND OUR DATASET

Causal reasoning is the process of determining how individual facts or conditions combine to produce an overall outcome (Penn & Povinelli, 2007). Although causal reasoning concerns understanding how mechanisms generate outcomes (Pearl, 2009), its deductive backbone ultimately reduces to patterns that can be expressed as logical implications; **logical reasoning**, therefore, captures the premise-conclusion structure that underlies many causal judgments Unlike simple associative prediction, it requires models to identify dependencies between propositions and to compute how local truths influence global conclusions. In practice, this almost always entails *multi-hop reasoning*, where intermediate inferences must be chained together across several steps. For example, given clauses $X$, $Y$, and $Z$, a model must first verify whether each clause holds (local checks), then com-

bine them through logical connectives (e.g., $X \vee Y$ at the clause level), and finally apply strict conjunctive control across all clauses to decide whether the full formula is satisfied (e.g., $(X \vee Y) \wedge Z$). This process highlights that accuracy depends not only on local classification but also on the correct *composition* of results across depth.

In the context of our work, we focus on testing specifically these requirements. By stratifying instances according to compositional depth, we enforce tasks where solving the problem requires multiple reasoning hops: at shallow depths, decisions can often be made with one or two local checks, but at greater depths, models must aggregate larger sets of clauses under global conjunctions. From our framing's perspective, reasoning over the structure (clause relations) is more important than the lexical relations encoded in them, and hence we make this the central aspect of our work.

### 3.2 Architectural Considerations for Logical Reasoning

In this section we provide an informal argument on how encoder layers could have an advantage over decoder-only architectures when dealing with logical reasoning.

Remark that logical classification requires aggregating dispersed evidence across an input sequence. Encoder architectures are well-suited to this task because each layer allows every token to integrate information from the entire sequence. This means that it will be able to express, in its own latent space, every element of an input sequence as a linear combination of the learned features and the other inputs. In other words, this allows for instant global information sharing.

Formally, let $s$ be an $n$-token input sequence $s = \langle x_1, x_2, \ldots x_n \rangle$, where every $x_i$ is represented by a $d_{\text{in}}$-dimensional vector; $x_i \in \mathbb{R}^{d_{\text{in}}}$. This sequence may be then rewritten as a matrix $X \in \mathbb{R}^{n \times d_{\text{in}}}$.

In the context of encoder layers, a encoder layer $\ell$ of hidden dimension $d$, for $\ell \colon \mathbb{R}^{n \times d_{\text{in}}} \to \mathbb{R}^{n \times d}$, transforms $X$ into a contextualized hidden state $H$. A classification decision would then be the result of pooling $h$ into a single vector $z = \text{pool}(h)$, for some pooling (aggregation) function $\text{pool} \colon \mathbb{R}^{n \times d} \to \mathbb{R}^d$. From this perspective, $z$ may be viewed as the output of a projection onto $\mathbb{R}^d$. Informally, this projection collapses the information from all tokens into a global representation. In other words, logical programs of the form

$$\text{(literals)} \ \Rightarrow \ \text{(clause-level disjunction)} \ \Rightarrow \ \text{(global conjunction)} \tag{1}$$

are encoded onto the projection. Given sufficient observations, this mechanism could allow a model to evaluate, in a single pass, such programs by repeatedly projecting and composing the *full* sequence.

In contrast, decoder-only architectures are recursive. To read and aggregate information distributed across the sequence, the model must propagate it step-by-step from left to right. There is still a projective step, but, algorithmically, the output at position $t$ depends only on the previously-observed and generated tokens. If the input clauses are not given in implication order, a solver will require some backtracking to fully consider and evaluate all given clauses. While reasoning models are able to do this to an extent through their "baked-in" chain-of-thought, it comes at the cost of a non-controllable inference process and multiple calls to the same model.

## 4 Methods

### 4.1 Dataset

From 3.1, it follows that a–comparatively–simple evaluation of causal reasoning capabilities could be carried out on FOL. Hence, we base our work on a benchmark known as SimpleLogic (Zhang et al., 2022). SimpleLogic is designed to evaluate deductive reasoning skills in a subset of FOL that excludes disjunctions. Every example from SimpleLogic is an algorithmically-generated tuple ⟨facts, rules, query, explanation, label⟩, where the facts are given atoms; the rules are definite clauses; the query is a single atom; and the label indicates whether the query can be deduced. All atoms are drawn from a vocabulary that leverages natural language (e.g., "Amy is sad"). The resulting entries may not be logical from a commonsense perspective, but they are valid within FOL. SimpleLogic uses a templatized language, which allows for controllable input length, linguistic variability, and reasoning depth–that is, the minimum number of reasoning steps needed to derive the

truth value of the query. We refer to the reasoning steps as "Proof Chain". In turn this ensures that difficulty is governed by logical complexity, rather than linguistic features.

In this work we create a base training set, and two OOD test datasets. The *training set* is akin to the original SimpleLogic work, with full natural-language strings generated by the base algorithm. It has 40,000 samples from depths 0 to 7, with 5,000 samples per depth. The **natural-language (NL) dataset** is a test set analogous to the training set, but manifests OOD by including deeper (up to 11) sequences. Finally, the **non-natural language (NNL) dataset** is constructed by sampling random characters to form an ungrammatical, likely unseen by the tokenizers, vocabulary; and then continue generating the dataset as before. Both test sets have 3,600 samples each, from depths 0 to 11, and 300 samples per depth. See Figure 2 for examples of entries in our corpora, and Appendix A for in-depth details.

```
Facts:  Batman is kind [PERIOD] Batman is generous        Facts:  Batman is a#d} [PERIOD] Batman is pqrs
Rules:  kind [AND] generous [IMPLY] helpful [PERIOD]      Rules:  a#d} [AND] y_hu] [IMPLY] u&^ho [PERIOD]
        helpful [IMPLY] friendly [PERIOD]                         u&^ho [IMPLY] {hu]? [PERIOD]
Query:  Batman is friendly [PERIOD]                       Query:  Batman is {hu]? [PERIOD]
Proof chain:                                              Proof chain:
     kind [AND] generous ⇒ helpful                            Cannot apply rule a#d} ∧ y_hu] ⇒ u&^ho
     helpful ⇒ friendly                                       because missing: y_hu]
Label: 1                                                  Label: 0
Depth: 2                                                  Depth: 1
```

Figure 2: Sample datapoints from our corpora. *Left*: an NL entry with depth 2. Remark that it will not always be a natural sentence, although it is guaranteed to be a valid set of clauses in SimpleLogic. The proof chain for this example contains the sequence of reasoning steps which solves it. *Right*: an NNL entry with depth 1. The proof chain in this example indicates that it is an unsolvable problem, given that the atom "y_hu]" is not in the derivation. In all corpora we use the separators [AND], [IMPLY], and [PERIOD] to separate elements of SimpleLogic from the atoms.

## 4.2 MODELS USED

For our encoder-only evaluation, we used two BERT variants (base and large), and for encoder-decoders we used BART variants (base and large). For the decoder-only models we evaluated two non-reasoning models, GPT-4.1 (OpenAI, 2025a) and Qwen 2.5 (Bai et al., 2025); and three reasoning models, GPT-5, Claude Opus 4.1 (Anthropic, 2025), and Qwen3-1.7B. All of these models are considered state-of-the-art LLMs, albeit only the Qwen-line of LLMs have open weights. We finetuned BERT, BART, and Qwen3. See Appendix B for further details on our methodology.

## 4.3 EVALUATION METRICS

We evaluate models using three complementary views: overall accuracy as a point estimate of correctness; *per–depth* precision, recall, and $F_1$-score to determine how performance changes with reasoning complexity; and threshold–swept discrimination via ROC curves and area under the curve (AUROC), which assesses ranking quality independent of a fixed decision threshold. We utilize the latter as our measure of statistical significance. Unless stated otherwise, depth–wise results are summarized with *macro* averages (the unweighted mean across depths) so that each depth contributes equally. Our setting is binary; accordingly, we report AUROC for the positive class. We describe our prompt methods, including how outputs are requested for each model family, in Appendix B.5. Given the nature of our experiments, we expect parsing errors in the LLMs across various settings. To mitigate this, we retry the calls up to five times; and to maintain consistency across data volumes, we default any failed calls to 0. We report analyses on the responses, including class distributions.

## 5 RESULTS

In this section we discuss the core results of our work on both corpora before and after finetuning (5.1 and 5.2, respectively). See 6 for ablation studies on these results.

## 5.1 NON-FINETUNED RESULTS

For our first experiment, we compared decoder-only models with the (unfinetuned) BERT, BART and Flan-T5 models. In this scenario–as expected–the BERT, BART and Flan-T5 models underpe-

formed. However, the LLMs had a more distinct behavioural pattern divided between the reasoning and non-reasoning types. For all models and datasets, we observed marginal changes (+0.5% average) in accuracy when increasing the number of shots from zero to five, and thus we only report zero-shot. In the NL dataset, the accuracies were 64% for GPT-4.1; 47% for Qwen-2.5; and 65% for Qwen-3 1.7B (zero and five-shot, respectively). In the NNL dataset, these were 65% for GPT-4.1; 53% for Qwen-2.5, and 61% for Qwen-3 1.7B. The API-based **reasoning models**, on the other hand, **had excellent performance**: GPT-5 achieved 100% accuracy in both the NL and NNL datasets. Claude Opus 4.1 scored 93% in the former (both zero and five-shot), and 65% and 66% in the latter (zero and five-shot, respectively).

Upon further inspection, we noted that the **low performances were *not* due to random guessing**, but, instead, the response patterns. In particular, for BART, BERT, Flan T5, and Qwen3-1.7B, we observed that the models often output the same label at every call (Figure 3) regardless of split.

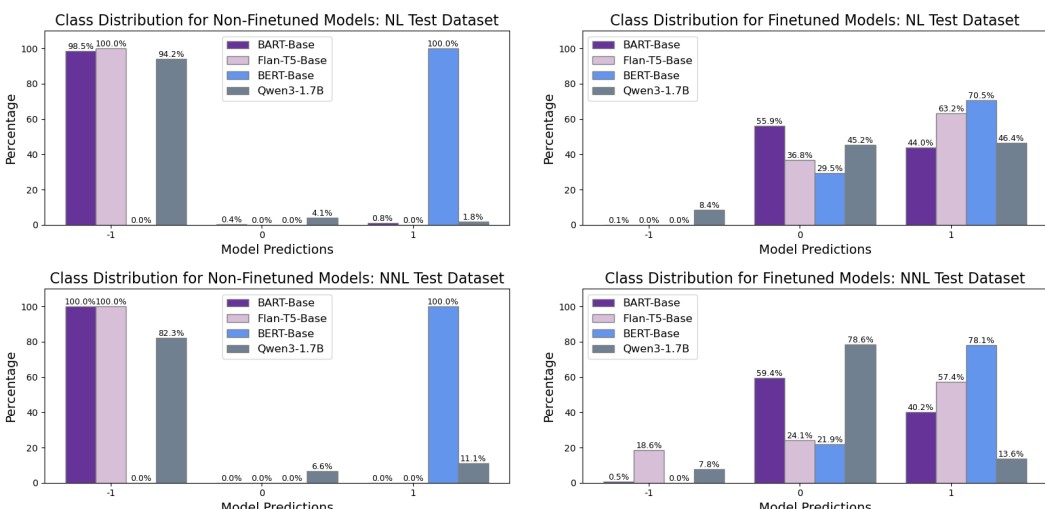

Figure 3: Class distribution for fine-tuned and non-finetuned models in the (*top*) NL and (*bottom*) NNL datasets. Parsing errors are marked as "-1". Fine-tuning improved the class distribution all across the board, with a less skewed distribution and fewer parsing errors. Label compliance in the NL dataset increased from 1.3% to 99.8% in BART-Base, and from 5.8% to 91.6% in Qwen3-1.7B. In the NNL dataset, label compliance increased from 0% to 91.2% in BART-Base, and from 17.7% to 92.2% in Qwen3-1.7B. Remark that in both cases there is still a label skewness in all models.

## 5.2 FINETUNED RESULTS

Next, we finetuned Flan-T5, BART and BERT, along with Qwen3-1.7B. **All models attained above-random, somewhat equivalent scores** in the NL split: 76% for Flan-T5 Base, 74% for BART-Base; 73% for Qwen3-1.7B; and 71% for BERT-Base. On the other hand, the NNL split proved to be more challenging: 61% for BERT-Base; 55% for BART-Base; 54% for Flan-T5 Base and 53% for Qwen3-1.7B.

Although these results would appear reasonable, observing **the AUC for all models revealed** that in the NL split, **Qwen-3 1.7B had near-random performance** (0.50), when compared to BART-Base, Flan-T5 Base and BERT-Base (0.62, 0.66, 0.76 and respectively), indicating better discrimination capabilities. In the NNL split, these numbers were 0.60, 0.53, 0.51 and 0.60 for Qwen-3 1.7B, BART-Base, Flan-T5 Base and BERT-Base, respectively. These were all improvements over the original, non-finetuned results, however: in the NL split the AUC increased by 0.06, 0.29, 0.38, and 0.08 (BART-Base, Flan-T5 Base, BERT-Base, Qwen-3 1.7B) and in NNL by 0.02, 0.07 0.09, and 0.1.

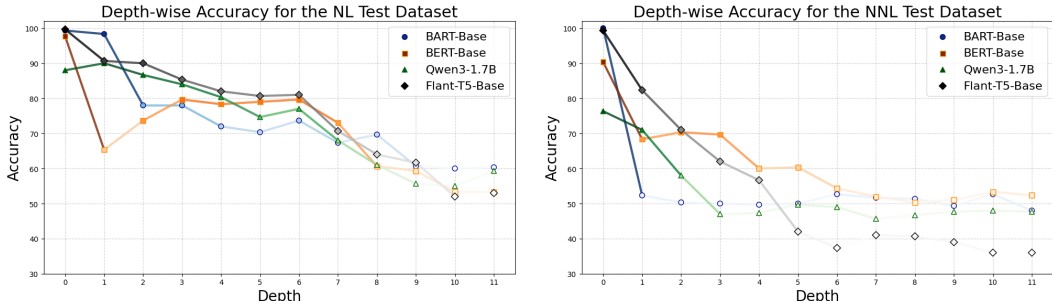

Figure 4: Averaged depth-wise accuracy for finetuned models in the NL (*left*) and NNL (*right*) datasets. Accuracy decreases with depth: in the NL dataset, average accuracy gradually declines from 90% to 50%; in the NNL dataset, it drops sharply from 89% to 50% and then plateaus. On average, encoder-based models outperformed decoder-only models across most depths.

## 6 ABLATION STUDIES

We present ablation studies on the relationship between the dataset depth and accuracy (6.1); and an evaluation on the inference time required to obtain our results (6.2).

### 6.1 ABLATION: (DATA) DEPTH VERSUS ACCURACY

In this study we decomposed the relationship between the number of reasoning steps and accuracy for the finetuned models only. Recall that both datasets manifested OOD by having deeper reasoning steps (8 to 11) than observed in the training data. The results are in Figure 4. For this we decomposed the predictions by depth. We observed a relatively even distribution on the prediction curves for the NL dataset. In the NNL dataset, however, the performance differences were more noticeable: **Flan-T5 Base and Qwen3-1.7B struggled to generalize to deeper reasoning chains**, rapidly reaching random guessing at depths 4 (Flan-T5 Base) and 3 (Qwen3-1.7B). On the other hand, BERT-Base was much slower to reach this state: depth 7. Numerically, we performed an OLS fit in the accuracy–depth profiles. Across models, the NL dataset exhibited a more gradual decline in accuracy (average slope = –3.45, std = 0.49), whereas the NNL dataset showed a less consistent but slightly flatter overall trend (average slope = –3.04, std = 1.31). We attribute this to the rapid accuracy drop in the NNL setting by depth 4, after which accuracy plateaus and flattens the global linear fit. To further isolate the initial region where performance deteriorates most, we repeated the OLS analysis over only the first four depths. In this regime, **the NNL dataset shows a substantially sharper decline** (average slope = –10.91, std = 3.36), compared to the NL dataset (average slope = –4.73, std = 2.45), confirming that accuracy degrades more abruptly in the absence of lexical cues.

### 6.2 ABLATION: INFERENCE TIME

We compared the accuracies of every model, along with the time and hardware requirements to acquire it. For this, we considered the *efficiency* of a model (accuracy over hours taken). We found GPT-5 to be the least efficient, in spite of its high performance, with an efficiency of 1.1: that is, it took over an hour to obtain a one-percent accuracy point. The most efficient was BART-Base; where one-percent accuracy points could be acquired every one tenth of a minute. Remark that LLMs took on average almost twice as long to compute the NNL split, due to the high amount of non-natural tokens produced. While we were unable to perform a full comparison given that the GPT and Claude models were behind an API–and hence the numbers above are estimates at best.

## 7 MECHANISTIC INTERPRETABILITY OF LOGICAL FLOW

Understanding how language models internally implement reasoning requires tools that go beyond input-output behavior. Recent theoretical work by Zhou et al. (2025) introduces a differential-geometric framework for analyzing the *flow of logic* inside hidden representations. Their key insight

| Model | Inference Time (hours; ↓) | Efficiency (Accuracy/Hour; ↑) | Hardware |
|---|---|---|---|
| BART-Base | 0.1 | 640 | Nvidia RTX 6000 |
| Flan-T5-Base | 0.45 | 143.4 | Nvidia RTX 6000 |
| BERT-Base | 0.17 | 388.2 | Nvidia RTX 6000 |
| Qwen2 | 1.08 | 43.52 | API |
| Qwen3-1.7B | 4.9 | 12.9 | Nvidia RTX 6000 |
| GPT-5 | 90.6 | 1.1 | API |
| GPT-4.1 | 0.5 | 129 | API |
| Claude Opus 4.1 | 14.4 | 5.5 | API |

Table 1: Average inference time in hours for zero-shot, averaged across both the NL and NNL datasets. Some calls were done through APIs, and so we are unable to provide accurate estimates comparing them. It is worth noting that the LLMs took on average twice as long on the NNL dataset, likely due to tokenization. To quantify the effort required to obtain the (average) accuracy, we provide the efficiency as a ratio of accuracy to time taken (in hours), with the highest (most efficient) model highlighted in blue (BART-Base); and the lowest in red (GPT-5). For BART-Base, Flan-T5-Base and Qwen3-1.7B, we consider the finetuned version of these models. These numbers are not fully comparable due to the (unknown) hardware used in some configurations.

is that reasoning manifests as a smooth trajectory in representation space, whose higher-order invariants, in particular, *curvature* encode the structural consistency of the underlying logical transformation. We build on this framework and adopt curvature similarity as our primary mechanistic probe, allowing us to evaluate whether different architectures maintain stable logical updates as reasoning depth increases.

**Curvature as a mechanistic probe.** Following Zhou et al. (2025), we interpret each incremental reasoning step as inducing a displacement in the model's hidden states. Curvature captures the second-order behavior of this trajectory and is invariant to superficial semantic variation. High curvature similarity across depths therefore indicates that the model re-applies a consistent internal update rule, rather than relying on shallow correlational shortcuts. This makes curvature a powerful, model-agnostic indicator of mechanistic structure in logical reasoning.

**Encoders preserve structural invariants.** Figure 5 shows depth-wise curvature similarity for four architectures. BERT (encoder-only) exhibits the highest and most stable similarity across depths 6–11, suggesting that its representations evolve according to a coherent geometric transformation. Encoder–decoder models (Flan-T5, Bart) partially preserve this behavior but degrade with depth, while the decoder-only Qwen displays the steepest decline. This mirrors the geometric findings of Zhou et al. (2025), where curvature structure collapses when the representation flow lacks global contextual integration.

**Mechanistic implication.** The ordering BERT > Flan-T5 > Bart > Qwen reflects the availability of *bidirectional contextualization* at representation time. Encoders expose every token to the full relational field, enabling the model to encode logical dependencies as a smooth, curvature-preserving flow. Decoder-only architectures, however, update representations autoregressively, accumulating local noise that disrupts higher-order invariants. Our results therefore provide mechanistic evidence that encoder components *substantially aid* the preservation of stable geometric transformations associated with logical reasoning, while decoder-only architectures exhibit greater curvature drift and reduced structural consistency.

## 8 CONCLUSION

While in-context learning (ICL) has propelled large language models to the forefront of AI research, our study shows that their ability to perform causal reasoning remains limited. We systematically compared encoder-based architectures with decoder-only LLMs under the hypothesis that the recursive nature of ICL is a hindrance rather than a benefit for reasoning over structured logical forms. Our results support this view: most LLMs—including state-of-the-art reasoning models—struggled to match the efficiency and robustness of encoder-only models such as BERT,

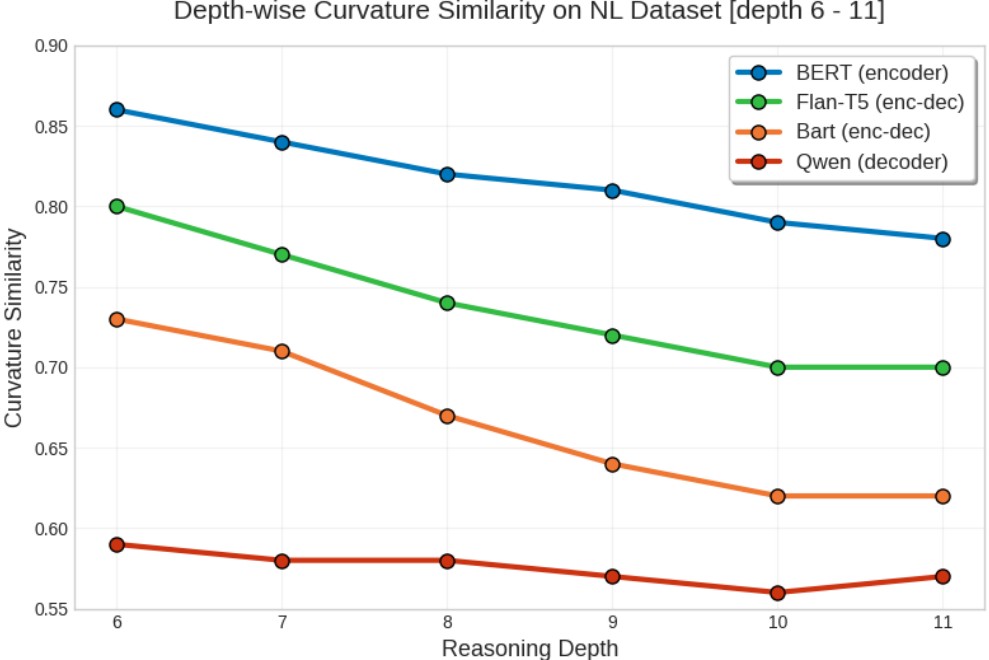

Figure 5: **Depth-wise curvature similarity across reasoning depths 6–11.** Encoder-only BERT maintains a stable geometric transformation across reasoning steps, reflecting strong preservation of logical invariants. Encoder–decoder models partially retain this structure, while the decoder-only Qwen shows pronounced curvature drift. These observations align with the geometric reasoning framework of Zhou et al. (2025).

particularly at greater reasoning depths and under lexical perturbations. Fine-tuned encoders and encoder–decoders demonstrated superior stability under distributional shifts, validating our mechanistic account of global projection versus recursive aggregation. The sole exception was GPT-5, which attained near-perfect accuracy across both natural-language and symbolic test sets. We hypothesize that this outlier performance stems from a combination of immense capacity and built-in chain-of-thought priors, though at the cost of substantially higher inference-time compute. Taken together, these findings suggest a practical trade-off: encoders and encoder–decoders remain the most resource-efficient and reliable choice for causal reasoning tasks, while decoder-only models can only close the gap at massive scale and cost.

Causal reasoning is important in many contemporary applications of LLMs, ranging from explainable AI to its applications to scientific discovery. Although LLMs are convenient and easy to use, the results shown here illustrate that their application must be done with caution. Our work also suggests that ICL is limited in its ability to properly capture causal compositionality. While further mathematical development is required to formally show the bounds and limits to which this occurs, empirical work could explore the development of architectures that merge the convenience of ICL with the capabilities of encoder-based architectures.

## 9 ETHICS

The datasets used for our experiments were generated synthetically for the specific purpose of evaluating logical reasoning and do not contain any personally identifiable information (PII) or sensitive user data. The language models evaluated, such as BERT, BART, Qwen, and GPT, were used in accordance with their intended research licenses. While the primary goal of this work is to advance the scientific understanding of AI reasoning, we acknowledge that enhancing logical capabilities in models could have dual-use applications. We encourage the responsible development and deployment of such technologies. The environmental impact associated with training and evaluating these

models was considered, and our findings highlight the efficiency of smaller, encoder-based models for specific tasks, which can guide more sustainable model selection in practice.

## 10    REPRODUCIBILITY STATEMENT

All code developed for data generation, model fine-tuning, and evaluation, along with the complete NL and NNL datasets, will be made publicly available in a GitHub repository upon publication. The pre-trained models used in this study are publicly available and were accessed from the Hugging Face Hub. Detailed configurations, including all hyperparameters, training scripts, and library versions (e.g., PyTorch, Transformers), will be provided in the repository to allow for the full replication of our experiments and to facilitate future research building upon this work. Hyperparameters, call parameters for API-based LLMs, and in-depth methodology are also reported in Appendix B

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

## A  DATASET GENERATION

We construct a supervised dataset of logical reasoning instances with grounded facts, Horn-style rules, and a target query. Each instance is automatically labeled as true or false ($\{0, 1\}$); and annotated with a minimal proof chain (or a failure trace) by running an algorithm based on Dijkstra's algorithm, which is available in the repository.

### A.1  PROBLEM SCHEMA AND DSL

Each example $\mathcal{E}$ comprises (i) a set of unary atoms (predicates) $\mathcal{A}$ over a single entity $e$ (e.g., `aggressive`, `uptight`), (ii) a set of Horn rules $\mathcal{R} = \{(\Pi_i \Rightarrow c_i)\}$ with $\Pi_i \subseteq \mathcal{A}$ and $c_i \in \mathcal{A}$, and (iii) a unary query $q \in \mathcal{A}$. We serialize $\mathcal{E}$ using a compact DSL:

$$\texttt{prem\_1 [AND] ...[AND] prem\_k [IMPLY] concl [PERIOD]}$$

We explicitly encode the logical connectives from SimpleLogic and the clause separator as tokens ([AND], [IMPLY], [PERIOD]) otherwise not present in $\mathcal{A}$. Facts are written as `entity is atom [PERIOD]` and queries as `query : entity is atom [PERIOD]`. This DSL maps deterministically to the internal graph representation described next.

### A.2  LABELING AND ANNOTATION

For each instance $\mathcal{E}$, we evaluate all premises in $\Pi \Rightarrow c$. If $q$ is reached, we backtrack to extract a minimal proof sequence (the proof chain), and otherwise we emit a structured failure trace that lists available premises (with their own proofs) and missing atoms.

**Depth Control and Curriculum**  We control reasoning depth by constraining the longest path length to the query. That is, $\text{depth}(q)$ is the minimum number of rule applications to derive $q$. We build a curriculum over depths $d \in \{1, \ldots, D_{\max}\}$, balancing the dataset across depths to prevent shallow-pattern overfitting.

**Negative Sampling Without Artifacts**  To avoid annotation shortcuts, we introduce three strategies: premise-missing negatives, where a random premise is removed from a $q$-concluding rule; distractor chains, where additional rules derive atoms unrelated to $q$; and adversarial swaps, where one premise is replaced with a semantically close predicate (e.g., `hot` vs. `uptight`) that already appears in the facts.

**Quality Checks**  We enforce the following invariants:

- **Well-formedness:** every rule has non-empty premises; all atoms occur in the vocabulary.
- **Sound labels:** recomputing our search algorithm reproduces $(y, \text{explanation})$ exactly.
- **Bounded depth:** $\text{depth}(q) \leq D_{\max}$.

Failed instances are discarded or repaired by re-sampling.

## B  DETAILED METHODS

### B.1  MODEL SPECIFICATIONS

For the decoder-only models we used GPT-4.1 (version SHORTCO-2025-04-14), GPT-5 (2025-08-07), Claude Opus 4.1 (20250805), Qwen-2.5 (vl-7B), Qwen3-17B, and Qwen3-1.7B. For encoder-only we used BERT (Large and Base). For encoder-decoder we used BART (Large and Base). Further details are in Table 2.

### B.2  CALL PARAMETERS

For the LLM calls, we set the temperature to zero whenever possible. The requested completion tokens were 128 for the non-reasoning models, and 5,000 tokens for the reasoning models. All our calls were made through the Azure OpenAI API.

| Model | Parameters | Type |
|---|---|---|
| GPT-4.1$^{\times}$ | - | Non-reasoning, ? |
| GPT-5$^{\times}$ | - | Reasoning, ? |
| Claude Opus 4.1$^{\times}$ | - | Reasoning, ? |
| Qwen-2.5 | 7B | Non-reasoning, decoder-only |
| Qwen3-17B | 17B | Reasoning, decoder-only |
| Qwen3-1.7B | 1.7B | Reasoning, decoder-only |
| BERT-Base | 110M | Non-reasoning,encoder-only |
| BART-Base | 139M | Non-reasoning,encoder-decoder |
| Flan-T5-Base | 250M | Non-reasoning,encoder-decoder |

Table 2: Models evaluated. For the models marked with $\times$, details regarding architecture, parameter size, or pretraining strategies have not been disclosed. We mark with ? models that we conjecture are decoder-only.

You are evaluating a subset of first-order logic.
In this subset, conjunctions are given by [AND], implications by [IMPLY], and separations between clauses as [PERIOD]
You will be given Facts, and Rules. Based on these, determine the truth value of the Query.
Your final answer should be 0 (if the Query is false) or 1 (if true).

Give your answer in JSON with the following schema:
{
"Label" (int): The label from the criterion. Only use the numbers 0 or 1.
}
Only use the key "Label".

Prompt 1: Prompt used for the LLMs. The data samples, as displayed in 2, are inserted as part of the user/assistant tuples in the case of five-shot. All LLMs typically adhered to the output format, and the parse failures mostly stemmed from API errors.

### B.3 FINETUNING

All models were finetuned for 3 epochs on a single NVIDIA RTX 6000 GPU with 48 GB of VRAM. A batch size of 8 was employed due to computational constraints, with a learning rate of $5 \times 10^{-5}$ yielding the best performance. We have finetuned explicitly with both components: the reasoning path (proof chain) and the final prediction. The models are therefore supervised not only on the direct label but also on the intermediate reasoning steps, ensuring that the evaluation reflects actual reasoning ability rather than surface-level label prediction.

### B.4 DATASET CREATION

Most of the dataset creation is covered in Appendix A. Throughout our generation, we fix random seeds at every stage and log the canonical form of each generated hypergraph, enabling exact regeneration of splits for future work.

### B.5 OUTPUT FORMAT

The prompt used for the LLMs is in Prompt 1.

### B.6 LEARNING CURVES

See Figure 6 for the finetuning loss for all modesl. For the same hyper-parameters, both Flan-T5 Base and BART-Base achieve a much smaller loss ($\approx 0.15$) whereas Qwen3-1.7B and BERT-Base converge to a loss value of $\approx 0.5$.

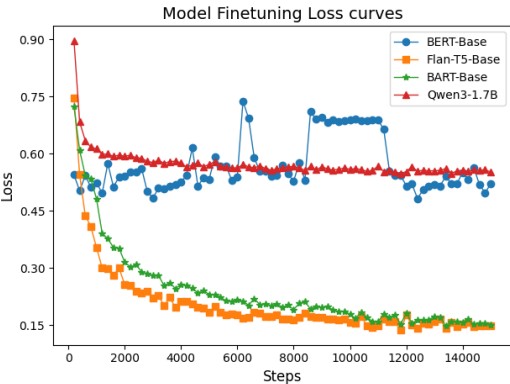

Figure 6: Finetuning loss curves for the models: Both BART-Base and Flan-T5 Base achieve better generalization.

## B.7 EVALUATING PROMPT SENSITIVITY

To assess the model's sensitivity to prompt formatting, we conducted automatic prompt optimization on GPT-4.1 Pryzant et al. (2023) using a beam size of 6 and a search depth of 8 on both dataset splits (See Table 5).

Table 5: Automatic prompt optimization on GPT 4.1. Marginal increase ($\approx 1\%$) in performance in comparison to evaluations without prompt optimization (Table 7 and 11)

| Dataset | Depth | **Metrics** | | | | | | |
|---------|-------|-----------|---------|---------|---------|---------|---------|----------|
| | | precision | | recall | | f1 | | accuracy |
| | | Label 0 | Label 1 | Label 0 | Label 1 | Label 0 | Label 1 | |
| NL | 0 | 00.00 | 100.0 | 00.00 | 100.0 | 00.00 | 100.0 | 100.0 |
| | 1 | 94.48 | 100.0 | 100.0 | 93.84 | 97.16 | 96.82 | 97.00 |
| | 2 | 73.30 | 92.55 | 95.57 | 61.27 | 82.97 | 73.73 | 79.33 |
| | 3 | 61.32 | 92.98 | 97.39 | 36.05 | 75.25 | 51.96 | 67.33 |
| | 4 | 52.57 | 85.11 | 95.00 | 25.00 | 67.68 | 38.65 | 57.67 |
| | 5 | 57.37 | 85.71 | 95.36 | 28.19 | 71.64 | 42.42 | 62.00 |
| | 6 | 54.09 | 86.05 | 95.86 | 23.87 | 69.15 | 37.37 | 58.67 |
| | 7 | 53.73 | 66.67 | 90.13 | 20.27 | 67.32 | 31.09 | 55.67 |
| | 8 | 56.86 | 60.00 | 88.96 | 19.71 | 69.38 | 29.67 | 57.33 |
| | 9 | 54.81 | 72.13 | 88.51 | 28.95 | 67.70 | 41.31 | 58.33 |
| | 10 | 52.94 | 44.44 | 84.38 | 14.29 | 65.06 | 21.62 | 51.67 |
| | 11 | 48.80 | 34.00 | 78.71 | 11.72 | 60.25 | 17.44 | 46.33 |
| | **Average** | 58.62 | 85.71 | 91.72 | 43.41 | 71.53 | 57.64 | 65.94 |
| NNL | 0 | 00.00 | 100.0 | 00.00 | 100.0 | 00.00 | 100.0 | 100.0 |
| | 1 | 93.07 | 93.94 | 94.00 | 93.00 | 93.53 | 93.47 | 93.50 |
| | 2 | 70.54 | 76.14 | 79.00 | 67.00 | 74.53 | 71.28 | 73.00 |
| | 3 | 58.65 | 59.38 | 61.00 | 57.00 | 59.80 | 58.16 | 59.00 |
| | 4 | 62.39 | 64.84 | 68.00 | 59.00 | 65.07 | 61.78 | 63.50 |
| | 5 | 59.05 | 60.00 | 62.00 | 57.00 | 60.49 | 58.46 | 59.50 |
| | 6 | 58.56 | 60.67 | 65.00 | 54.00 | 61.61 | 57.14 | 59.50 |
| | 7 | 58.62 | 56.64 | 51.00 | 64.00 | 54.55 | 60.09 | 57.50 |
| | 8 | 57.30 | 55.86 | 51.00 | 62.00 | 53.97 | 58.77 | 56.50 |
| | 9 | 57.33 | 54.40 | 43.00 | 68.00 | 49.14 | 60.44 | 55.50 |
| | 10 | 51.47 | 50.76 | 35.00 | 67.00 | 41.67 | 57.76 | 51.00 |
| | 11 | 60.27 | 55.91 | 44.00 | 71.00 | 50.87 | 62.56 | 57.50 |
| | **Average** | 63.15 | 67.28 | 59.36 | 70.69 | 61.20 | 68.94 | 65.50 |

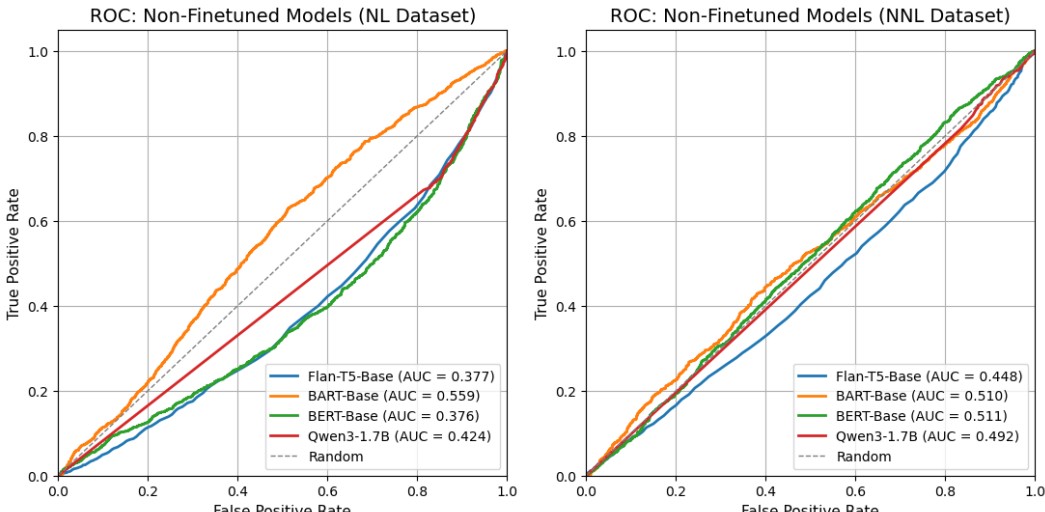

Figure 7: ROC curves for non-finetuned models. Performance is near random across all models in both datasets.

## C    AUC/ROC EVALUATION

### C.1    NON-FINETUNED

In the NL dataset (Appendix C.1), the AUC values for all models (Flant-T5 Base, BART-Base, BERT-Base, and Qwen3-1.7B) are near 0.5 (ranging from 0.4 to 0.6). The ROC curves closely follow the dashed random line, confirming that the models' performance on this task, without fine-tuning, is near random across the board. In the NNL dataset (Appendix C.1), the ROC curves for Flan-T5 Base (AUC 0.448), BART-Base (r. 0.5), BERT-Base (r. 0.5), and Qwen3-1.7B (r. 0.5) are also nearly indistinguishable from the random baseline, and, as before, the performance of the non-finetuned models in this corpus is random

### C.2    FINETUNED

In the NL dataset, the AUC values of all models except Qwen3-1.7B are above random- $0.66, 0.62$ and $0.76$ for Flan-T5 Base, BART-Base and BERT-Base respectively. The ROC curves, as shown in Appendix C.2 are above the dashed random line, illustrating strong discrimination capabilities for encoder based architectures. In the NNL dataset the ROC curves for Flan-T5 Base (AUC 0.513), BART-Base (r. 0.53), BERT-Base (r. 0.6) and Qwen3-1.7B (r. 0.6) are above random but the AUC gains smaller compared to the NL dataset.

## D    TEST RESULTS ON THE NATURAL LANGUAGE DATASET

We evaluate four finetuned models—BERT-Base, BART-large, Flan-T5-Base and Qwen-3-1.7B—on the Natural Language benchmark across depths 0–11 (See Figure 11, Table 6). Averaged over depths, Flan-T5-Base achieves the highest accuracy (0.76), followed by BART-Base (0.74) and Qwen-3-1.7B (0.73) and . Per-class analysis shows that Flan-T5-base generally balances precision and recall across both labels, BART-Base has high precision and recall for Label 1 and 0 respectively, while Qwen-3-1.7B excels in F1 for Label 0, and BERT-Base combines strong precision on Label 0 with high recall on Label 1.

We also evaluate current reasoning and non-reasoning models (See Figure 10, Tables 7 to 9). GPT-5, Claude Opus 4 (Zero and five shot) have very strong performance across all depths, with GPT-5 achieving near perfect accuracy across all depths. The non-reasoning models (GPT-4 Qwen 2.5, Qwen3-17B) struggle to generalize at higher depths. GPT 4.1 (zero and five shot) and Qwen3-17B

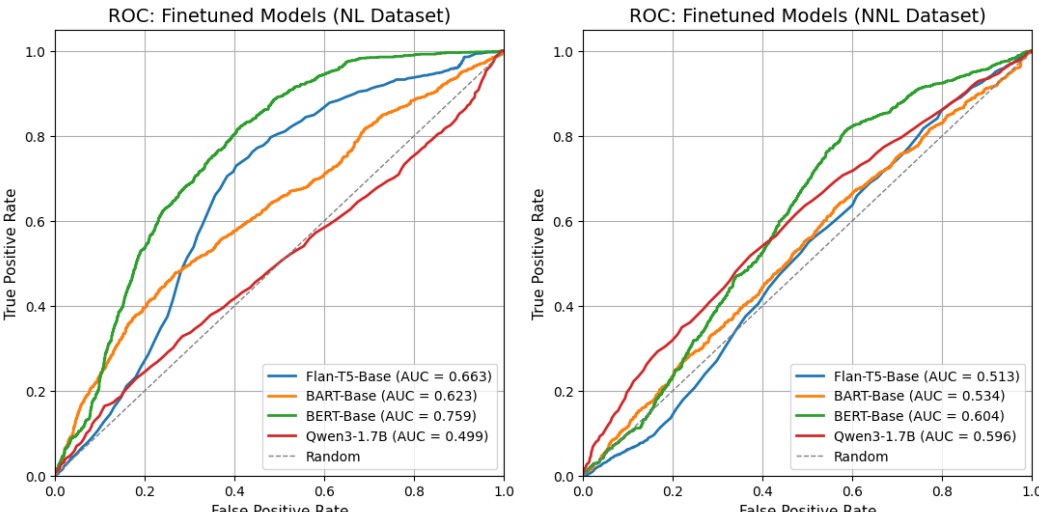

Figure 8: ROC curves for finetuned models. Finetuning markedly boosts discrimination on the NL dataset. The models illustrate strong cross-domain generalization in the NNL dataset with smaller absolute gains compared to the NL dataset.

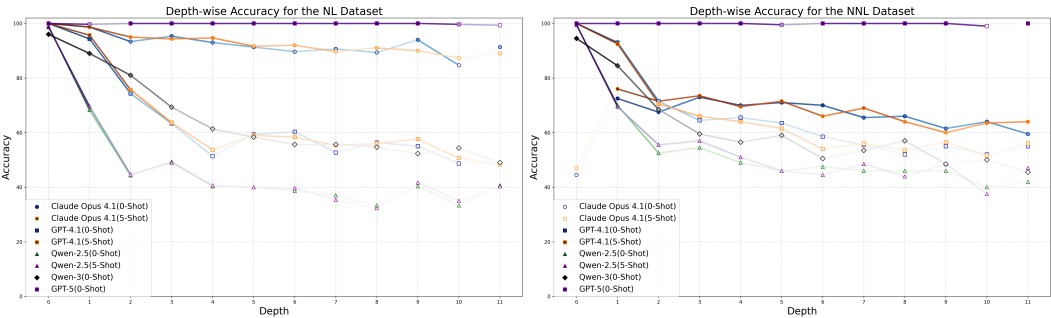

Figure 9: Averaged depth-wise accuracy for Decoder-only models in the NL (*left*) and NNL (*right*) datasets. In general, accuracy decreases with depth: in the NL dataset, average accuracy gradually declines from 90% to 50%; in the NNL dataset, it drops sharply from 89% to 50% and then plateaus.

(Zero shot) have roughly similar accuracy ($\approx 0.65$). Qwen 2.5 (zero and five shot) performs worse than average ($0.47$)(Table 9).

# E    TEST RESULTS ON THE NON-NATURAL LANGUAGE DATASET

On the Non-Natural Language benchmark, overall accuracies are lower than in the Natural Language setting for the finetuned models. BERT-Base performing best (0.61), followed by BART-base (0.55), Flan-T5-Base (0.54) and Qwen-3-1.7B (0.53) (See Figure 13). Per-class results reveal complementary strengths: BERT-base balances Label 0 precision and Label 1 recall/F1, Qwen-3-1.7B strongly favors Label 0 (high recall/F1) but suffers from poor Label 1 recall, while Flan-T5-Base and BART-base remain intermediate across metrics.

Performance on the non-natural language test set varies considerably across models.GPT-5 achieves near-perfect accuracy (Figure 12, Table 11), clearly outperforming others. Claude Opus 4.1 attains 0.66 accuracy in the five-shot setting and 0.65 in zero-shot (Table 12), marking a substantial drop compared to its performance on the natural language test dataset. GPT-4.1, by contrast, maintains consistent accuracy (0.65–0.66) across both test datasets. The Qwen family shows a different trend

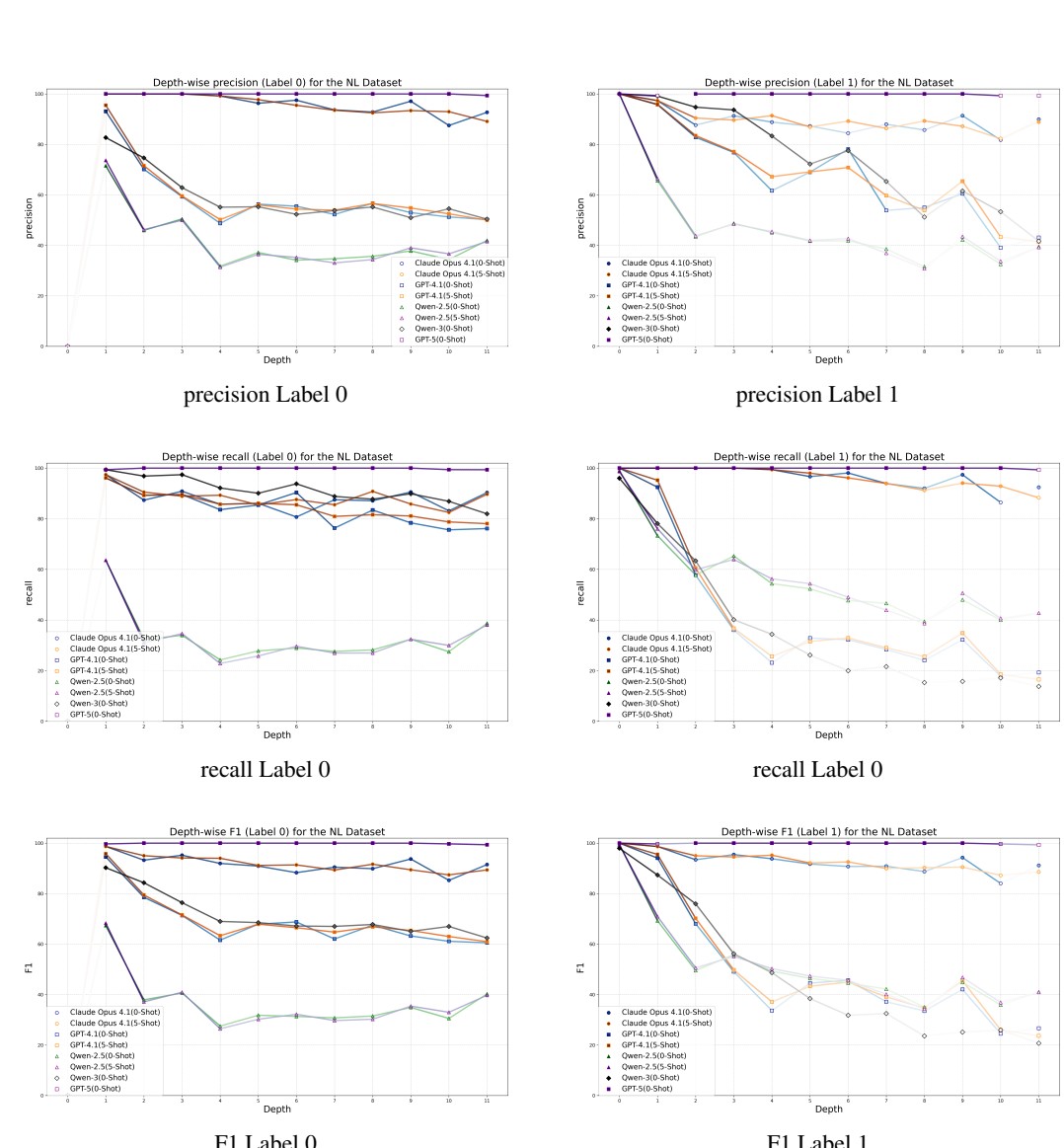

precision Label 0                    precision Label 1

recall Label 0                       recall Label 0

F1 Label 0                           F1 Label 1

Figure 10: Depth-wise metrics of the Decoder only models for the NL Dataset

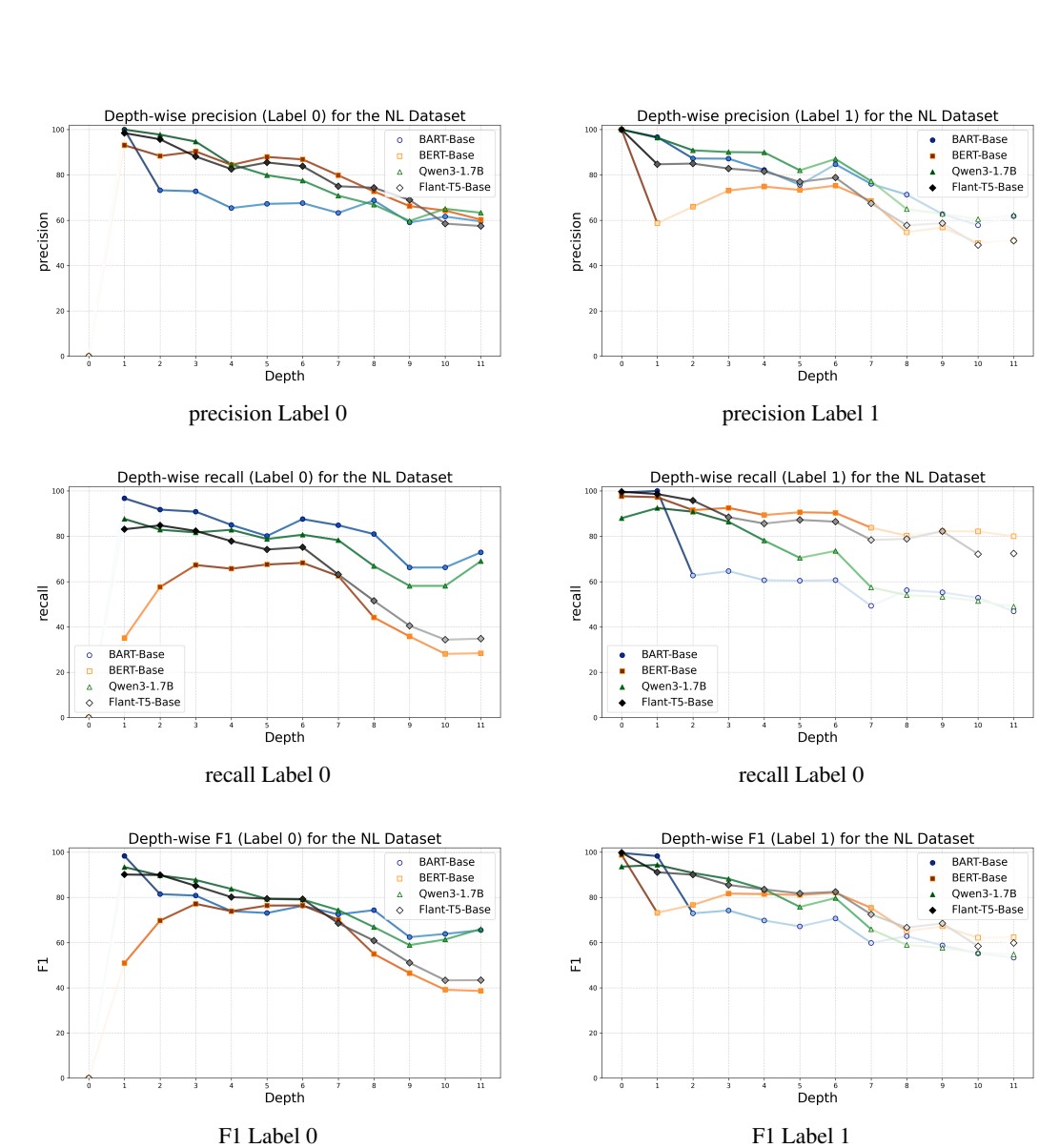

Figure 11: Depth-wise metrics of the finetuned models for the NL Dataset

Table 6: Performance of finetuned models on Natural Language Depth 12 test data. We highlight the best (**bold**) and second-best (underline) values.

| Model | Depth | Metrics | | | | | | |
|---|---|---|---|---|---|---|---|---|
| | | precision | | recall | | $F_1$ | | accuracy |
| | | Label 0 | Label 1 | Label 0 | Label 1 | Label 0 | Label 1 | |
| BERT-Base | 0 | 0.000 | 100.00 | 0.000 | 97.67 | 0.000 | 98.82 | 97.67 |
| | 1 | 93.10 | 58.68 | 35.06 | 97.26 | 50.94 | 73.20 | 65.33 |
| | 2 | 88.35 | 65.99 | 57.59 | 91.55 | 69.73 | 76.70 | 73.67 |
| | 3 | 90.35 | 73.12 | 67.32 | 92.52 | 77.15 | 81.68 | 79.67 |
| | 4 | 84.40 | 74.87 | 65.71 | 89.38 | 73.90 | 81.48 | 78.33 |
| | 5 | 87.93 | 73.37 | 67.55 | 90.60 | 76.40 | 81.08 | 79.00 |
| | 6 | 86.84 | 75.27 | 68.28 | 90.32 | 76.45 | 82.11 | 79.67 |
| | 7 | 79.83 | 68.51 | 62.50 | 83.78 | 70.11 | 75.38 | 73.00 |
| | 8 | 72.73 | 54.73 | 44.17 | 80.29 | 54.96 | 65.09 | 60.67 |
| | 9 | 66.25 | 56.82 | 35.81 | 82.24 | 46.49 | 67.20 | 59.33 |
| | 10 | 64.29 | 50.00 | 28.12 | 82.14 | 39.13 | 62.16 | 53.33 |
| | 11 | 60.27 | 51.10 | 28.39 | 80.00 | 38.60 | 62.37 | 53.33 |
| | **Average** | 80.00 | 67.34 | 50.63 | **89.00** | 62.00 | 76.65 | 71.00 |
| BART-Base | 0 | 0.00 | 100.00 | 0.00 | 99.33 | 0.00 | 99.67 | 99.33 |
| | 1 | 100.00 | 96.69 | 96.75 | 100.00 | 98.35 | 98.32 | 98.33 |
| | 2 | 73.23 | 87.25 | 91.77 | 62.68 | 81.46 | 72.95 | 78.00 |
| | 3 | 72.77 | 87.16 | 90.85 | 64.63 | 80.81 | 74.22 | 78.00 |
| | 4 | 65.38 | 82.20 | 85.00 | 60.62 | 73.91 | 69.78 | 72.00 |
| | 5 | 67.22 | 75.63 | 80.13 | 60.40 | 73.11 | 67.16 | 70.33 |
| | 6 | 67.55 | 84.68 | 87.59 | 60.65 | 76.28 | 70.68 | 73.67 |
| | 7 | 63.24 | 76.04 | 84.87 | 49.32 | 72.47 | 59.84 | 67.33 |
| | 8 | 68.75 | 71.30 | 80.98 | 56.20 | 74.37 | 62.86 | 69.67 |
| | 9 | 59.04 | 62.69 | 66.22 | 55.26 | 62.42 | 58.74 | 60.67 |
| | 10 | 61.63 | 57.81 | 66.25 | 52.86 | 63.86 | 55.22 | 60.00 |
| | 11 | 59.47 | 61.82 | 72.90 | 46.90 | 65.51 | 53.33 | 60.33 |
| | **Average** | 68.42 | **81.00** | **82.00** | 66.89 | 75.00 | 73.32 | 73.00 |
| Flan-T5 Base | 0 | 00.00 | 100.00 | 00.00 | 99.67 | 00.00 | 99.83 | 99.67 |
| | 1 | 98.46 | 84.71 | 83.12 | 98.63 | 90.14 | 91.14 | 90.67 |
| | 2 | 95.71 | 85.00 | 84.81 | 95.77 | 89.93 | 90.07 | 90.00 |
| | 3 | 88.11 | 82.80 | 82.35 | 88.44 | 85.14 | 85.53 | 85.33 |
| | 4 | 82.58 | 81.55 | 77.86 | 85.62 | 80.15 | 83.54 | 82.00 |
| | 5 | 85.50 | 76.92 | 74.17 | 87.25 | 79.43 | 81.76 | 80.67 |
| | 6 | 83.85 | 78.82 | 75.17 | 86.45 | 79.27 | 82.46 | 81.00 |
| | 7 | 75.00 | 67.44 | 63.16 | 78.38 | 68.57 | 72.50 | 70.67 |
| | 8 | 74.34 | 57.75 | 51.53 | 78.83 | 60.87 | 66.67 | 64.00 |
| | 9 | 68.97 | 58.69 | 40.54 | 82.24 | 51.06 | 68.49 | 61.67 |
| | 10 | 58.51 | 49.03 | 34.38 | 72.14 | 43.31 | 58.38 | 52.00 |
| | 11 | 57.45 | 50.97 | 34.84 | 72.41 | 43.37 | 59.83 | 53.00 |
| | **Average** | **81** | 73.12 | 63.55 | 87 | 71.09 | **80** | **76** |
| Qwen-3-1.7B | 0 | 00.00 | 100.00 | 00.00 | 88.00 | 00.00 | 93.62 | 88.00 |
| | 1 | 100.00 | 96.43 | 87.66 | 92.47 | 93.43 | 94.41 | 90.00 |
| | 2 | 97.76 | 90.85 | 82.91 | 90.85 | 89.73 | 90.85 | 86.67 |
| | 3 | 94.70 | 90.07 | 81.70 | 86.39 | 87.72 | 88.19 | 84.00 |
| | 4 | 84.67 | 89.93 | 82.86 | 78.12 | 83.75 | 83.61 | 80.33 |
| | 5 | 79.87 | 82.03 | 78.81 | 70.47 | 79.33 | 75.81 | 74.67 |
| | 6 | 77.48 | 87.02 | 80.69 | 73.55 | 79.05 | 79.72 | 77.00 |
| | 7 | 70.83 | 77.27 | 78.29 | 57.43 | 74.38 | 65.89 | 68.00 |
| | 8 | 66.87 | 64.91 | 66.87 | 54.01 | 66.87 | 58.96 | 61.00 |
| | 9 | 59.72 | 62.79 | 58.11 | 53.29 | 58.90 | 57.65 | 55.67 |
| | 10 | 65.03 | 60.50 | 58.13 | 51.43 | 61.39 | 55.60 | 55.00 |
| | 11 | 63.31 | 62.28 | 69.03 | 48.97 | 66.05 | 54.83 | 59.33 |
| | **Average** | 77.31 | **83** | 75 | 71.94 | **76** | 77 | 73.31 |

Table 7: Performance of GPT models on Natural Language Depth 12 test data. We highlight the best (**bold**) and second-best (underline) values. All numeric values are rounded to two decimal places.

| Model | Depth | Metrics | | | | | | |
|---|---|---|---|---|---|---|---|---|
| | | precision | | recall | | $F_1$ | | accuracy |
| | | Label 0 | Label 1 | Label 0 | Label 1 | Label 0 | Label 1 | |
| GPT 5 (Zero Shot) | 0 | 00.00 | 100.0 | 00.00 | 100.0 | 00.00 | 100.0 | 100.0 |
| | 1 | 100.0 | 99.32 | 99.35 | 100.0 | 99.67 | 99.66 | 99.67 |
| | 2 | 100.0 | 100.0 | 100.0 | 100.0 | 100.0 | 100.0 | 100.0 |
| | 3 | 100.0 | 100.0 | 100.0 | 100.0 | 100.0 | 100.0 | 100.0 |
| | 4 | 100.0 | 100.0 | 100.0 | 100.0 | 100.0 | 100.0 | 100.0 |
| | 5 | 100.0 | 100.0 | 100.0 | 100.0 | 100.0 | 100.0 | 100.0 |
| | 6 | 100.0 | 100.0 | 100.0 | 100.0 | 100.0 | 100.0 | 100.0 |
| | 7 | 100.0 | 100.0 | 100.0 | 100.0 | 100.0 | 100.0 | 100.0 |
| | 8 | 100.0 | 100.0 | 100.0 | 100.0 | 100.0 | 100.0 | 100.0 |
| | 9 | 100.0 | 100.0 | 100.0 | 100.0 | 100.0 | 100.0 | 100.0 |
| | 10 | 100.0 | 99.29 | 99.38 | 100.0 | 99.69 | 99.64 | 99.67 |
| | 11 | 99.35 | 99.31 | 99.35 | 99.31 | 99.35 | 99.31 | 99.33 |
| | **Average** | **99.94** | **99.84** | **99.82** | **99.95** | **99.88** | **99.90** | **99.89** |
| GPT 4.1 (Five Shot) | 0 | 00.00 | 100.0 | 00.00 | 100.0 | 00.00 | 100.0 | 100.0 |
| | 1 | 95.48 | 95.86 | 96.10 | 95.21 | 95.79 | 95.53 | 95.67 |
| | 2 | 71.57 | 83.50 | 89.24 | 60.56 | 79.44 | 70.20 | 75.67 |
| | 3 | 59.57 | 77.14 | 89.54 | 36.73 | 71.54 | 49.77 | 63.67 |
| | 4 | 50.21 | 67.21 | 85.71 | 25.62 | 63.32 | 37.10 | 53.67 |
| | 5 | 56.03 | 69.12 | 86.09 | 31.54 | 67.89 | 43.32 | 59.00 |
| | 6 | 54.39 | 70.83 | 85.52 | 32.90 | 66.49 | 44.93 | 58.33 |
| | 7 | 53.95 | 59.72 | 80.92 | 29.05 | 64.74 | 39.09 | 55.33 |
| | 8 | 56.60 | 53.85 | 81.60 | 25.55 | 66.83 | 34.65 | 56.00 |
| | 9 | 54.79 | 65.43 | 81.08 | 34.87 | 65.40 | 45.49 | 57.67 |
| | 10 | 52.50 | 43.33 | 78.75 | 18.57 | 63.00 | 26.00 | 50.67 |
| | 11 | 50.00 | 41.38 | 78.06 | 16.55 | 60.96 | 23.65 | 48.33 |
| | **Average** | 58.20 | 77.84 | 84.75 | 46.80 | 69.01 | 58.45 | 64.50 |
| GPT 4.1 (Zero Shot) | 0 | 00.00 | 100.0 | 00.00 | 100.0 | 00.00 | 100.0 | 100.0 |
| | 1 | 93.08 | 95.74 | 96.10 | 92.47 | 94.57 | 94.08 | 94.33 |
| | 2 | 70.15 | 82.83 | 89.24 | 57.75 | 78.55 | 68.05 | 74.33 |
| | 3 | 59.31 | 76.81 | 89.54 | 36.05 | 71.35 | 49.07 | 63.33 |
| | 4 | 48.75 | 61.67 | 83.57 | 23.13 | 61.58 | 33.64 | 51.33 |
| | 5 | 56.33 | 69.01 | 85.43 | 32.89 | 67.89 | 44.55 | 59.33 |
| | 6 | 55.51 | 78.12 | 90.34 | 32.26 | 68.77 | 45.66 | 60.33 |
| | 7 | 52.25 | 53.85 | 76.32 | 28.38 | 62.03 | 37.17 | 52.67 |
| | 8 | 56.67 | 55.00 | 83.44 | 24.09 | 67.49 | 33.50 | 56.33 |
| | 9 | 52.97 | 60.49 | 78.38 | 32.24 | 63.22 | 42.06 | 55.00 |
| | 10 | 51.27 | 39.06 | 75.62 | 17.86 | 61.11 | 24.51 | 48.67 |
| | 11 | 50.21 | 43.08 | 76.13 | 19.31 | 60.51 | 26.67 | 48.67 |
| | **Average** | 57.60 | 76.65 | 83.98 | 45.97 | 68.33 | 57.47 | 63.69 |

Table 8: Performance of Claude Opus 4.1 models on Natural Language Depth 12 test data. We highlight the best (**bold**) and second-best (underline) values. All numeric values are rounded to two decimal places.

| Model | Depth | Metrics | | | | | | |
| --- | --- | --- | --- | --- | --- | --- | --- | --- |
| | | precision | | recall | | $F_1$ | | accuracy |
| | | Label 0 | Label 1 | Label 0 | Label 1 | Label 0 | Label 1 | |
| Claude Opus 4.1 (Five shot) | 0 | 00.00 | 100.0 | 00.00 | 100.0 | 00.00 | 100.0 | 100.0 |
| | 1 | 100.0 | 97.33 | 97.40 | 100.0 | 98.68 | 98.65 | 98.67 |
| | 2 | 100.0 | 90.45 | 90.51 | 100.0 | 95.02 | 94.98 | 95.00 |
| | 3 | 100.0 | 89.63 | 88.89 | 100.0 | 94.12 | 94.53 | 94.33 |
| | 4 | 99.21 | 91.38 | 89.29 | 99.38 | 93.98 | 95.21 | 94.67 |
| | 5 | 97.73 | 86.90 | 85.43 | 97.99 | 91.17 | 92.11 | 91.67 |
| | 6 | 95.49 | 89.22 | 87.59 | 96.13 | 91.37 | 92.55 | 92.00 |
| | 7 | 93.53 | 86.34 | 85.53 | 93.92 | 89.35 | 89.97 | 89.67 |
| | 8 | 92.50 | 89.29 | 90.80 | 91.24 | 91.64 | 90.25 | 91.00 |
| | 9 | 93.38 | 87.20 | 85.81 | 94.08 | 89.44 | 90.51 | 90.00 |
| | 10 | 92.96 | 82.28 | 82.50 | 92.86 | 87.42 | 87.25 | 87.33 |
| | 11 | 89.10 | 88.89 | 89.68 | 88.28 | 89.39 | 88.58 | 89.00 |
| | **Average** | 95.69 | **90.57** | **88.51** | 96.51 | **91.96** | **93.45** | **92.78** |
| Claude Opus 4.1 (Zero shot) | 0 | 00.00 | 100.0 | 00.00 | 100.0 | 00.00 | 100.0 | 100.0 |
| | 1 | 100.0 | 97.33 | 97.40 | 100.0 | 98.68 | 98.65 | 98.67 |
| | 2 | 100.0 | 87.65 | 87.34 | 100.0 | 93.24 | 93.42 | 93.33 |
| | 3 | 100.0 | 91.30 | 90.85 | 100.0 | 95.21 | 95.45 | 95.33 |
| | 4 | 99.17 | 88.83 | 85.71 | 99.38 | 91.95 | 93.81 | 93.00 |
| | 5 | 96.30 | 87.27 | 86.09 | 96.64 | 90.91 | 91.72 | 91.33 |
| | 6 | 97.50 | 84.44 | 80.69 | 98.06 | 88.30 | 90.75 | 89.67 |
| | 7 | 93.66 | 87.97 | 87.50 | 93.92 | 90.48 | 90.85 | 90.67 |
| | 8 | 92.81 | 85.71 | 87.12 | 91.97 | 89.87 | 88.73 | 89.33 |
| | 9 | 97.10 | 91.36 | 90.54 | 97.37 | 93.71 | 94.27 | 94.00 |
| | 10 | 87.50 | 81.76 | 83.13 | 86.43 | 85.26 | 84.03 | 84.67 |
| | 11 | 92.72 | 89.93 | 90.32 | 92.41 | 91.50 | 91.16 | 91.33 |
| | **Average** | **95.91** | 90.15 | 87.91 | **96.72** | 91.73 | 93.32 | 92.61 |

(Table 13): Qwen 2.5 (zero and five shots) improves by $6\%$ over its performance on the natural language test dataset while Qwen3-17B shows performs slightly worse (0.61).

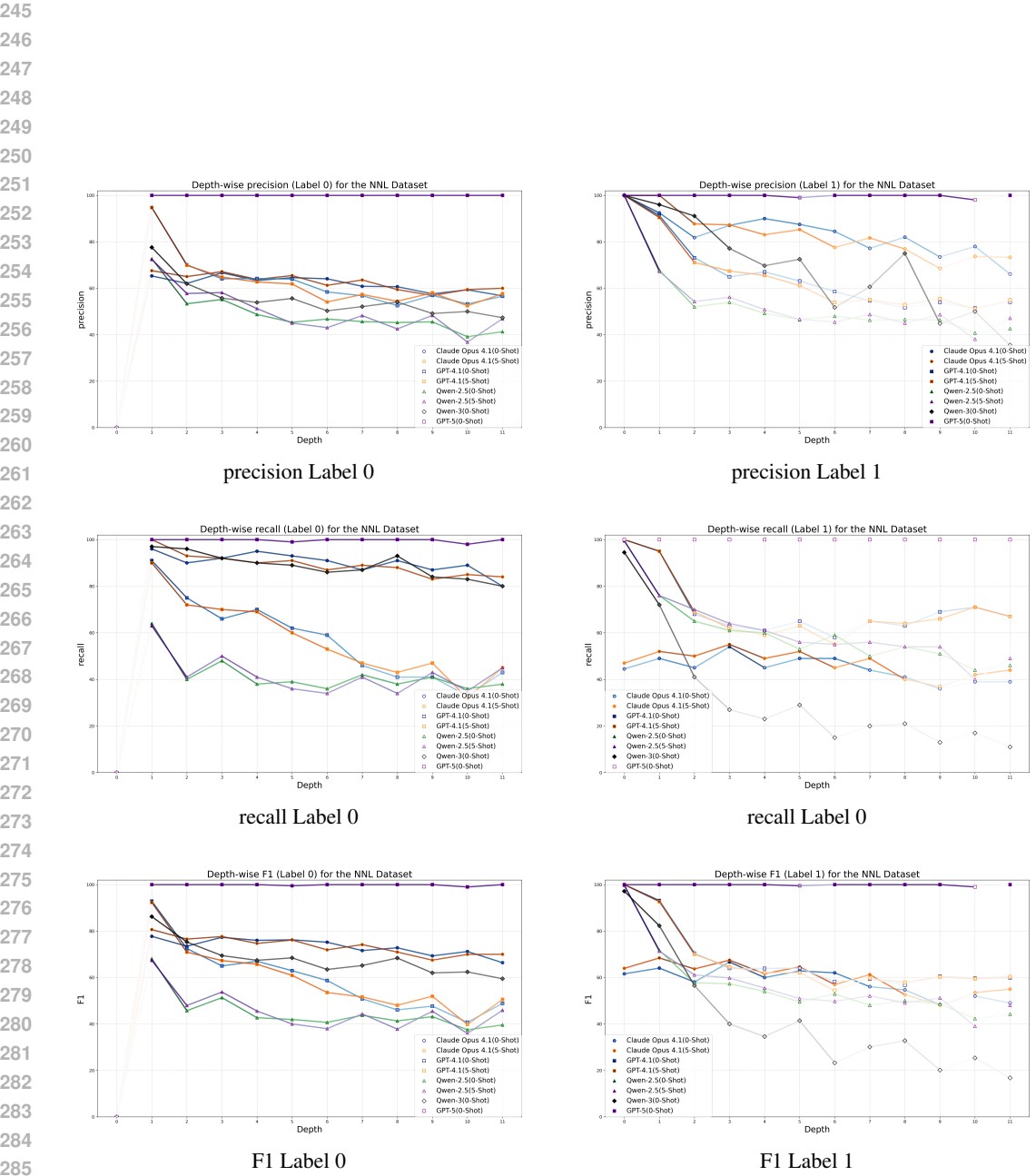

Figure 12: Depth-wise metrics of the Decoder only models for the NNL Dataset

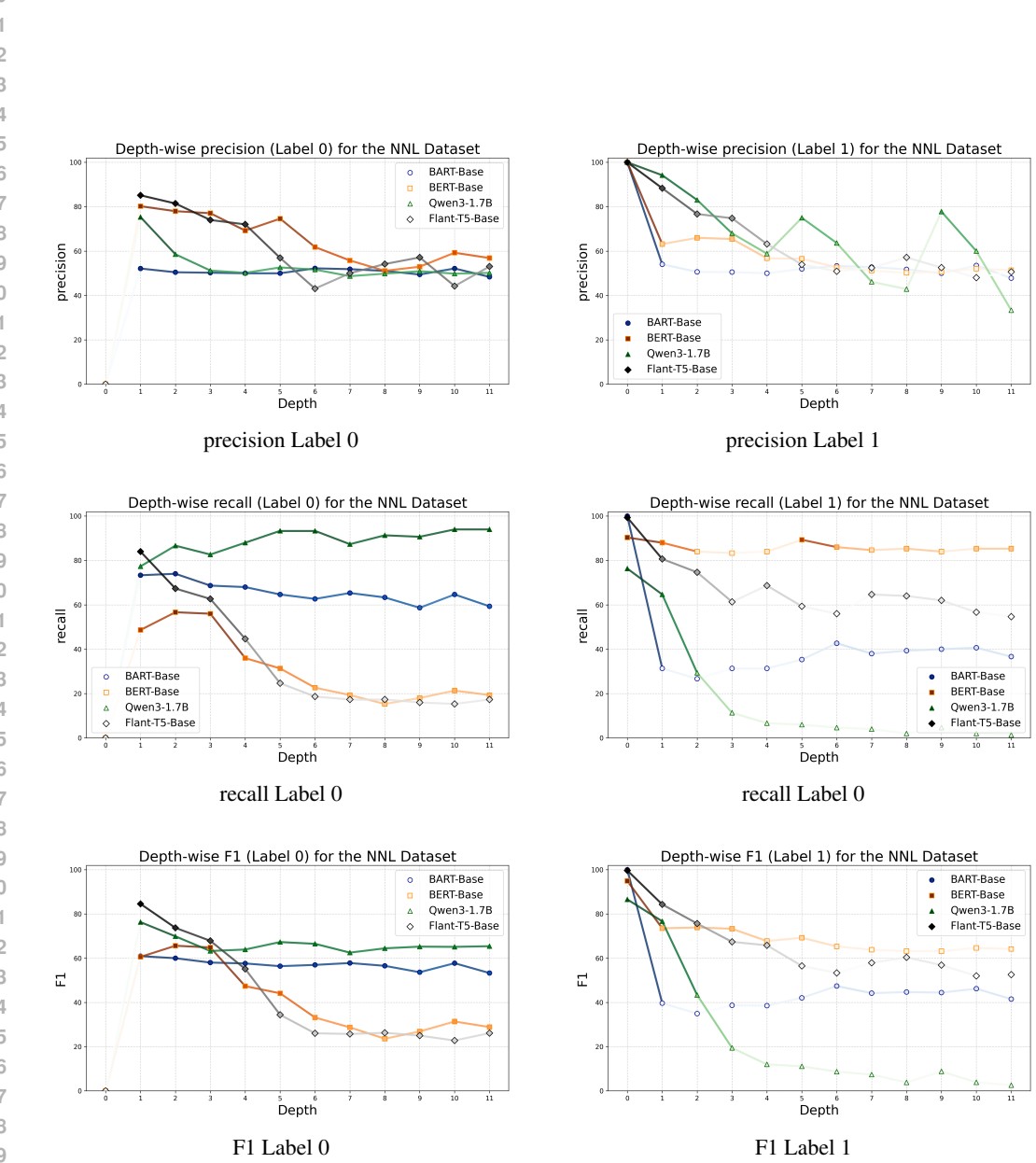

Figure 13: Depth-wise metrics of the finetuned only models for the NNL Dataset

Table 9: Performance of Qwen 2.5 models on Natural Language Depth 12 test data. We highlight the best (**bold**) and second-best (underline) values. All numeric values are rounded to two decimal places.

| Model | Depth | **Metrics** | | | | | | |
|---|---|---|---|---|---|---|---|---|
| | | precision | | recall | | $F_1$ | | accuracy |
| | | Label 0 | Label 1 | Label 0 | Label 1 | Label 0 | Label 1 | |
| Qwen 2.5 (Five Shot) | 0 | 00.00 | 100.0 | 00.00 | 98.67 | 00.00 | 99.33 | 98.67 |
| | 1 | 73.68 | 66.47 | 63.64 | 76.03 | 68.29 | 70.93 | 69.67 |
| | 2 | 46.23 | 43.81 | 31.01 | 59.86 | 37.12 | 50.60 | 44.67 |
| | 3 | 50.00 | 48.45 | 34.64 | 63.95 | 40.93 | 55.13 | 49.00 |
| | 4 | 31.37 | 45.45 | 22.86 | 56.25 | 26.45 | 50.28 | 40.67 |
| | 5 | 36.45 | 41.97 | 25.83 | 54.36 | 30.23 | 47.37 | 40.00 |
| | 6 | 35.25 | 42.70 | 29.66 | 49.03 | 32.21 | 45.65 | 39.67 |
| | 7 | 33.06 | 36.93 | 26.97 | 43.92 | 29.71 | 40.12 | 35.33 |
| | 8 | 34.38 | 30.81 | 26.99 | 38.69 | 30.24 | 34.30 | 32.33 |
| | 9 | 39.02 | 43.50 | 32.43 | 50.66 | 35.42 | 46.81 | 41.67 |
| | 10 | 36.64 | 33.73 | 30.00 | 40.71 | 32.99 | 36.89 | 35.00 |
| | 11 | 41.55 | 39.24 | 38.06 | 42.76 | 39.73 | 40.92 | 40.33 |
| | **Average** | 41.72 | 50.48 | 33.00 | **59.71** | 36.85 | 54.71 | 47.25 |
| Qwen 2.5 (Zero Shot) | 0 | 00.00 | 100.0 | 00.00 | 99.00 | 00.00 | 99.50 | 99.00 |
| | 1 | 71.53 | 65.64 | 63.64 | 73.29 | 67.35 | 69.26 | 68.33 |
| | 2 | 45.95 | 43.39 | 32.28 | 57.75 | 37.92 | 49.55 | 44.33 |
| | 3 | 50.49 | 48.73 | 33.99 | 65.31 | 40.62 | 55.81 | 49.33 |
| | 4 | 31.78 | 45.08 | 24.29 | 54.37 | 27.53 | 49.29 | 40.33 |
| | 5 | 37.17 | 41.71 | 27.81 | 52.35 | 31.82 | 46.43 | 40.00 |
| | 6 | 34.15 | 41.81 | 28.97 | 47.74 | 31.34 | 44.58 | 38.67 |
| | 7 | 34.71 | 38.55 | 27.63 | 46.62 | 30.77 | 42.20 | 37.00 |
| | 8 | 35.66 | 31.58 | 28.22 | 39.42 | 31.51 | 35.06 | 33.33 |
| | 9 | 37.80 | 42.20 | 32.43 | 48.03 | 34.91 | 44.92 | 40.33 |
| | 10 | 34.38 | 32.56 | 27.50 | 40.00 | 30.56 | 35.90 | 33.33 |
| | 11 | 41.96 | 39.49 | 38.71 | 42.76 | 40.27 | 41.06 | 40.67 |
| | **Average** | 41.56 | 50.33 | 33.29 | 59.08 | 36.97 | 54.36 | 47.06 |
| Qwen3-17B (Zero Shot) | 0 | 00.00 | 100.0 | 00.00 | 96.00 | 00.00 | 97.96 | 96.00 |
| | 1 | 82.70 | 99.13 | 99.35 | 78.08 | 90.27 | 87.36 | 89.00 |
| | 2 | 74.63 | 94.74 | 96.84 | 63.38 | 84.30 | 75.95 | 81.00 |
| | 3 | 62.87 | 93.65 | 97.39 | 40.14 | 76.41 | 56.19 | 69.33 |
| | 4 | 55.13 | 83.33 | 92.14 | 34.38 | 68.98 | 48.67 | 61.33 |
| | 5 | 55.28 | 72.22 | 90.07 | 26.17 | 68.51 | 38.42 | 58.33 |
| | 6 | 52.31 | 77.50 | 93.79 | 20.00 | 67.16 | 31.79 | 55.67 |
| | 7 | 53.78 | 65.31 | 88.82 | 21.62 | 67.00 | 32.49 | 55.67 |
| | 8 | 55.21 | 51.22 | 87.73 | 15.33 | 67.77 | 23.60 | 54.67 |
| | 9 | 50.96 | 61.54 | 89.86 | 15.79 | 65.04 | 25.13 | 52.33 |
| | 10 | 54.51 | 53.33 | 86.88 | 17.14 | 66.99 | 25.95 | 54.33 |
| | 11 | 50.40 | 41.67 | 81.94 | 13.79 | 62.41 | 20.73 | 49.00 |
| | **Average** | **57.70** | **84.52** | **91.30** | 41.49 | **70.71** | **55.66** | **64.72** |

Table 10: Performance of finetuned models on Non-Natural Language Depth 12 test data. We highlight the best (**bold**) and second-best (underline) values. All numeric values are rounded to two decimal places.

| Model | Depth | Metrics | | | | | | |
|---|---|---|---|---|---|---|---|---|
| | | precision | | recall | | $F_1$ | | accuracy |
| | | Label 0 | Label 1 | Label 0 | Label 1 | Label 0 | Label 1 | |
| BERT-base | 0 | 00.00 | 100.0 | 00.00 | 90.33 | 00.00 | 94.92 | 90.33 |
| | 1 | 80.22 | 63.16 | 48.67 | 88.00 | 60.58 | 73.54 | 68.33 |
| | 2 | 77.98 | 65.97 | 56.67 | 84.00 | 65.64 | 73.90 | 70.33 |
| | 3 | 77.06 | 65.45 | 56.00 | 83.33 | 64.86 | 73.31 | 69.67 |
| | 4 | 69.23 | 56.76 | 36.00 | 84.00 | 47.37 | 67.74 | 60.00 |
| | 5 | 74.60 | 56.54 | 31.33 | 89.33 | 44.13 | 69.25 | 60.33 |
| | 6 | 61.82 | 52.65 | 22.67 | 86.00 | 33.17 | 65.32 | 54.33 |
| | 7 | 55.77 | 51.21 | 19.33 | 84.67 | 28.71 | 63.82 | 52.00 |
| | 8 | 51.11 | 50.20 | 15.33 | 85.33 | 23.59 | 63.21 | 50.33 |
| | 9 | 52.94 | 50.60 | 18.00 | 84.00 | 26.87 | 63.16 | 51.00 |
| | 10 | 59.26 | 52.03 | 21.33 | 85.33 | 31.37 | 64.65 | 53.33 |
| | 11 | 56.86 | 51.41 | 19.33 | 85.33 | 28.86 | 64.16 | 52.33 |
| | **Average** | **65.69** | 59.72 | 31.33 | **86.15** | 42.43 | **70.54** | **61.03** |
| BART-base | 0 | 00.00 | 100.0 | 00.00 | 100.0 | 00.00 | 100.0 | 100.0 |
| | 1 | 52.13 | 54.02 | 73.33 | 31.33 | 60.94 | 39.66 | 52.33 |
| | 2 | 50.45 | 50.63 | 74.00 | 26.67 | 60.00 | 34.93 | 50.33 |
| | 3 | 50.24 | 50.54 | 68.67 | 31.33 | 58.03 | 38.68 | 50.00 |
| | 4 | 50.00 | 50.00 | 68.00 | 31.33 | 57.63 | 38.52 | 49.67 |
| | 5 | 50.00 | 51.96 | 64.67 | 35.33 | 56.40 | 42.06 | 50.00 |
| | 6 | 52.22 | 53.33 | 62.67 | 42.67 | 56.97 | 47.41 | 52.67 |
| | 7 | 51.85 | 52.78 | 65.33 | 38.00 | 57.82 | 44.19 | 51.67 |
| | 8 | 51.08 | 51.75 | 63.33 | 39.33 | 56.55 | 44.70 | 51.33 |
| | 9 | 49.44 | 50.00 | 58.67 | 40.00 | 53.66 | 44.44 | 49.33 |
| | 10 | 52.15 | 53.51 | 64.67 | 40.67 | 57.74 | 46.21 | 52.67 |
| | 11 | 48.37 | 47.83 | 59.33 | 36.67 | 53.29 | 41.51 | 48.00 |
| | **Average** | 50.73 | 61.55 | 65.70 | 45.64 | 57.25 | 52.41 | 54.83 |
| Flan-T5-Base | 0 | 00.00 | 100.0 | 00.00 | 99.33 | 00.00 | 99.67 | 99.33 |
| | 1 | 85.14 | 88.32 | 84.00 | 80.67 | 84.56 | 84.32 | 82.33 |
| | 2 | 81.45 | 76.71 | 67.33 | 74.67 | 73.72 | 75.68 | 71.00 |
| | 3 | 74.02 | 74.80 | 62.67 | 61.33 | 67.87 | 67.40 | 62.00 |
| | 4 | 72.04 | 63.19 | 44.67 | 68.67 | 55.14 | 65.81 | 56.67 |
| | 5 | 56.92 | 53.94 | 24.67 | 59.33 | 34.42 | 56.51 | 42.00 |
| | 6 | 43.08 | 50.91 | 18.67 | 56.00 | 26.05 | 53.33 | 37.33 |
| | 7 | 50.00 | 52.43 | 17.33 | 64.67 | 25.74 | 57.91 | 41.00 |
| | 8 | 54.17 | 57.14 | 17.33 | 64.00 | 26.26 | 60.38 | 40.67 |
| | 9 | 57.14 | 52.54 | 16.00 | 62.00 | 25.00 | 56.88 | 39.00 |
| | 10 | 44.23 | 48.02 | 15.33 | 56.67 | 22.77 | 51.99 | 36.00 |
| | 11 | 53.06 | 50.62 | 17.33 | 54.67 | 26.13 | 52.56 | 36.00 |
| | **Average** | **66.74** | 65.44 | 35.03 | 69.33 | 45.95 | 67.33 | 53.61 |
| Qwen-3-1.7B | 0 | 00.00 | 100.0 | 00.00 | 76.33 | 00.00 | 86.58 | 76.33 |
| | 1 | 75.32 | 94.17 | 77.33 | 64.67 | 76.32 | 76.68 | 71.00 |
| | 2 | 58.56 | 83.02 | 86.67 | 29.33 | 69.89 | 43.35 | 58.00 |
| | 3 | 51.24 | 68.00 | 82.67 | 11.33 | 63.27 | 19.43 | 47.00 |
| | 4 | 50.19 | 58.82 | 88.00 | 06.67 | 63.92 | 11.98 | 47.33 |
| | 5 | 52.63 | 75.00 | 93.33 | 06.00 | 67.31 | 11.11 | 49.67 |
| | 6 | 51.66 | 63.64 | 93.33 | 04.67 | 66.51 | 08.70 | 49.00 |
| | 7 | 48.70 | 46.15 | 87.33 | 04.00 | 62.53 | 07.36 | 45.67 |
| | 8 | 49.82 | 42.86 | 91.33 | 02.00 | 64.47 | 03.82 | 46.67 |
| | 9 | 50.94 | 77.78 | 90.67 | 04.67 | 65.23 | 08.81 | 47.67 |
| | 10 | 49.82 | 60.00 | 94.00 | 02.00 | 65.13 | 03.87 | 48.00 |
| | 11 | 50.18 | 33.33 | 94.00 | 01.33 | 65.43 | 02.56 | 47.67 |
| | **Average** | 51.89 | **88.57** | **88.97** | 22.26 | **65.66** | 35.57 | 52.83 |

Table 11: Performance of GPT models on Non-Natural Language Depth 12 test data. We highlight the best (**bold**) and second-best (underline) values. All numeric values are rounded to two decimal places.

| Model | Depth | Metrics | | | | | | |
| --- | --- | --- | --- | --- | --- | --- | --- | --- |
| | | precision | | recall | | $F_1$ | | accuracy |
| | | Label 0 | Label 1 | Label 0 | Label 1 | Label 0 | Label 1 | |
| GPT 5 (Zero Shot) | 0 | 00.00 | 100.0 | 00.00 | 100.0 | 00.00 | 100.0 | 100.0 |
| | 1 | 100.0 | 100.0 | 100.0 | 100.0 | 100.0 | 100.0 | 100.0 |
| | 2 | 100.0 | 100.0 | 100.0 | 100.0 | 100.0 | 100.0 | 100.0 |
| | 3 | 100.0 | 100.0 | 100.0 | 100.0 | 100.0 | 100.0 | 100.0 |
| | 4 | 100.0 | 100.0 | 100.0 | 100.0 | 100.0 | 100.0 | 100.0 |
| | 5 | 100.0 | 99.01 | 99.00 | 100.0 | 99.50 | 99.50 | 99.50 |
| | 6 | 100.0 | 100.0 | 100.0 | 100.0 | 100.0 | 100.0 | 100.0 |
| | 7 | 100.0 | 100.0 | 100.0 | 100.0 | 100.0 | 100.0 | 100.0 |
| | 8 | 100.0 | 100.0 | 100.0 | 100.0 | 100.0 | 100.0 | 100.0 |
| | 9 | 100.0 | 100.0 | 100.0 | 100.0 | 100.0 | 100.0 | 100.0 |
| | 10 | 100.0 | 98.04 | 98.00 | 100.0 | 98.99 | 99.01 | 99.00 |
| | 11 | 100.0 | 100.0 | 100.0 | 100.0 | 100.0 | 100.0 | 100.0 |
| | **Average** | **100.0** | **99.77** | **99.73** | **100.0** | **99.86** | **99.88** | **99.88** |
| GPT 4.1 (Five Shot) | 0 | 00.00 | 100.0 | 00.00 | 100.0 | 00.00 | 100.0 | 100.0 |
| | 1 | 94.74 | 90.48 | 90.00 | 95.00 | 92.31 | 92.68 | 92.50 |
| | 2 | 69.90 | 71.13 | 72.00 | 69.00 | 70.94 | 70.05 | 70.50 |
| | 3 | 64.81 | 67.39 | 70.00 | 62.00 | 67.31 | 64.58 | 66.00 |
| | 4 | 62.73 | 65.56 | 69.00 | 59.00 | 65.71 | 62.11 | 64.00 |
| | 5 | 61.86 | 61.17 | 60.00 | 63.00 | 60.91 | 62.07 | 61.50 |
| | 6 | 54.08 | 53.92 | 53.00 | 55.00 | 53.54 | 54.46 | 54.00 |
| | 7 | 57.32 | 55.08 | 47.00 | 65.00 | 51.65 | 59.63 | 56.00 |
| | 8 | 54.43 | 52.89 | 43.00 | 64.00 | 48.04 | 57.92 | 53.50 |
| | 9 | 58.02 | 55.46 | 47.00 | 66.00 | 51.93 | 60.27 | 56.50 |
| | 10 | 52.46 | 51.08 | 32.00 | 71.00 | 39.75 | 59.41 | 51.50 |
| | 11 | 57.69 | 54.92 | 45.00 | 67.00 | 50.56 | 60.36 | 56.00 |
| | **Average** | 63.31 | 66.48 | 57.09 | 72.00 | 60.04 | 69.13 | 65.17 |
| GPT 4.1 (Zero Shot) | 0 | 00.00 | 100.0 | 00.00 | 100.0 | 00.00 | 100.0 | 100.0 |
| | 1 | 94.79 | 91.35 | 91.00 | 95.00 | 92.86 | 93.14 | 93.00 |
| | 2 | 70.09 | 73.12 | 75.00 | 68.00 | 72.46 | 70.47 | 71.50 |
| | 3 | 64.08 | 64.95 | 66.00 | 63.00 | 65.02 | 63.96 | 64.50 |
| | 4 | 64.22 | 67.03 | 70.00 | 61.00 | 66.99 | 63.87 | 65.50 |
| | 5 | 63.92 | 63.11 | 62.00 | 65.00 | 62.94 | 64.04 | 63.50 |
| | 6 | 58.42 | 58.59 | 59.00 | 58.00 | 58.71 | 58.29 | 58.50 |
| | 7 | 56.79 | 54.62 | 46.00 | 65.00 | 50.83 | 59.36 | 55.50 |
| | 8 | 52.56 | 51.64 | 41.00 | 63.00 | 46.07 | 56.76 | 52.00 |
| | 9 | 56.94 | 53.91 | 41.00 | 69.00 | 47.67 | 60.53 | 55.00 |
| | 10 | 53.23 | 51.45 | 33.00 | 71.00 | 40.74 | 59.66 | 52.00 |
| | 11 | 56.58 | 54.03 | 43.00 | 67.00 | 48.86 | 59.82 | 55.00 |
| | **Average** | 63.85 | 66.64 | 57.00 | 72.69 | 60.23 | 69.54 | 65.50 |

Table 12: Performance of Claude Opus 4.1 model on Non-Natural Language Depth 12 test data. We highlight the best (**bold**) and second-best (underline) values. All numeric values are rounded to two decimal places.

| Model | Depth | Metrics | | | | | | |
| --- | --- | --- | --- | --- | --- | --- | --- | --- |
| | | precision | | recall | | $F_1$ | | accuracy |
| | | Label 0 | Label 1 | Label 0 | Label 1 | Label 0 | Label 1 | |
| Claude Opus 4.1 (Five shot) | 0 | 00.00 | 100.0 | 00.00 | 47.00 | 00.00 | 63.95 | 47.00 |
| | 1 | 67.57 | 100.0 | 100.0 | 52.00 | 80.65 | 68.42 | 76.00 |
| | 2 | 65.03 | 87.72 | 93.00 | 50.00 | 76.54 | 63.69 | 71.50 |
| | 3 | 67.15 | 87.30 | 92.00 | 55.00 | 77.64 | 67.48 | 73.50 |
| | 4 | 63.83 | 83.05 | 90.00 | 49.00 | 74.69 | 61.64 | 69.50 |
| | 5 | 65.47 | 85.25 | 91.00 | 52.00 | 76.15 | 64.60 | 71.50 |
| | 6 | 61.27 | 77.59 | 87.00 | 45.00 | 71.90 | 56.96 | 66.00 |
| | 7 | 63.57 | 81.67 | 89.00 | 49.00 | 74.17 | 61.25 | 69.00 |
| | 8 | 59.46 | 76.92 | 88.00 | 40.00 | 70.97 | 52.63 | 64.00 |
| | 9 | 56.85 | 68.52 | 83.00 | 37.00 | 67.48 | 48.05 | 60.00 |
| | 10 | 59.44 | 73.68 | 85.00 | 42.00 | 69.96 | 53.50 | 63.50 |
| | 11 | 60.00 | 73.33 | 84.00 | 44.00 | 70.00 | 55.00 | 64.00 |
| | **Average** | **58.70** | 83.77 | 89.27 | **46.85** | **70.83** | **60.09** | **66.29** |
| Claude Opus 4.1 (Zero shot) | 0 | 00.00 | 100.0 | 00.00 | 44.50 | 00.00 | 61.59 | 44.50 |
| | 1 | 65.31 | 92.45 | 96.00 | 49.00 | 77.73 | 64.05 | 72.50 |
| | 2 | 62.07 | 81.82 | 90.00 | 45.00 | 73.47 | 58.06 | 67.50 |
| | 3 | 66.67 | 87.10 | 92.00 | 54.00 | 77.31 | 66.67 | 73.00 |
| | 4 | 63.33 | 90.00 | 95.00 | 45.00 | 76.00 | 60.00 | 70.00 |
| | 5 | 64.58 | 87.50 | 93.00 | 49.00 | 76.23 | 62.82 | 71.00 |
| | 6 | 64.08 | 84.48 | 91.00 | 49.00 | 75.21 | 62.03 | 70.00 |
| | 7 | 60.84 | 77.19 | 87.00 | 44.00 | 71.60 | 56.05 | 65.50 |
| | 8 | 60.67 | 82.00 | 91.00 | 41.00 | 72.80 | 54.67 | 66.00 |
| | 9 | 57.62 | 73.47 | 87.00 | 36.00 | 69.32 | 48.32 | 61.50 |
| | 10 | 59.33 | 78.00 | 89.00 | 39.00 | 71.20 | 52.00 | 64.00 |
| | 11 | 56.74 | 66.10 | 80.00 | 39.00 | 66.39 | 49.06 | 59.50 |
| | **Average** | 57.89 | **84.16** | **90.09** | 44.54 | 70.48 | 58.25 | 65.42 |

Table 13: Performance of Qwen 2.5 models on Non-Natural Language Depth 12 test data. We highlight the best (**bold**) and second-best (underline) values. All numeric values are rounded to two decimal places.

| Model | Depth | Metrics | | | | | | |
|---|---|---|---|---|---|---|---|---|
| | | precision | | recall | | $F_1$ | | accuracy |
| | | Label 0 | Label 1 | Label 0 | Label 1 | Label 0 | Label 1 | |
| Qwen 2.5 (Five Shot) | 0 | 00.00 | 100.0 | 00.00 | 99.50 | 00.00 | 99.75 | 99.50 |
| | 1 | 72.41 | 67.26 | 63.00 | 76.00 | 67.38 | 71.36 | 69.50 |
| | 2 | 57.75 | 54.26 | 41.00 | 70.00 | 47.95 | 61.14 | 55.50 |
| | 3 | 58.14 | 56.14 | 50.00 | 64.00 | 53.76 | 59.81 | 57.00 |
| | 4 | 51.25 | 50.83 | 41.00 | 61.00 | 45.56 | 55.45 | 51.00 |
| | 5 | 45.00 | 46.67 | 36.00 | 56.00 | 40.00 | 50.91 | 46.00 |
| | 6 | 43.04 | 45.45 | 34.00 | 55.00 | 37.99 | 49.77 | 44.50 |
| | 7 | 48.24 | 48.70 | 41.00 | 56.00 | 44.32 | 52.09 | 48.50 |
| | 8 | 42.50 | 45.00 | 34.00 | 54.00 | 37.78 | 49.09 | 44.00 |
| | 9 | 48.31 | 48.65 | 43.00 | 54.00 | 45.50 | 51.18 | 48.50 |
| | 10 | 36.84 | 38.10 | 35.00 | 40.00 | 35.90 | 39.02 | 37.50 |
| | 11 | 46.88 | 47.12 | 45.00 | 49.00 | 45.92 | 48.04 | 47.00 |
| | **Average** | 49.84 | 56.70 | 42.09 | **64.15** | 45.64 | **60.19** | 54.04 |
| Qwen 2.5 (Zero Shot) | 0 | 00.00 | 100.0 | 00.00 | 99.50 | 00.00 | 99.75 | 99.50 |
| | 1 | 72.73 | 67.86 | 64.00 | 76.00 | 68.09 | 71.70 | 70.00 |
| | 2 | 53.33 | 52.00 | 40.00 | 65.00 | 45.71 | 57.78 | 52.50 |
| | 3 | 55.17 | 53.98 | 48.00 | 61.00 | 51.34 | 57.28 | 54.50 |
| | 4 | 48.72 | 49.18 | 38.00 | 60.00 | 42.70 | 54.05 | 49.00 |
| | 5 | 45.35 | 46.49 | 39.00 | 53.00 | 41.94 | 49.53 | 46.00 |
| | 6 | 46.75 | 47.97 | 36.00 | 59.00 | 40.68 | 52.91 | 47.50 |
| | 7 | 45.65 | 46.30 | 42.00 | 50.00 | 43.75 | 48.08 | 46.00 |
| | 8 | 45.24 | 46.55 | 38.00 | 54.00 | 41.30 | 50.00 | 46.00 |
| | 9 | 45.56 | 46.36 | 41.00 | 51.00 | 43.16 | 48.57 | 46.00 |
| | 10 | 39.13 | 40.74 | 36.00 | 44.00 | 37.50 | 42.31 | 40.00 |
| | 11 | 41.30 | 42.59 | 38.00 | 46.00 | 39.58 | 44.23 | 42.00 |
| | **Average** | 48.83 | 56.10 | 41.82 | 62.92 | 45.05 | 59.32 | 53.25 |
| Qwen3-17B (Zero Shot) | 0 | 00.00 | 100.0 | 00.00 | 94.50 | 00.00 | 97.17 | 94.50 |
| | 1 | 77.60 | 96.00 | 97.00 | 72.00 | 86.22 | 82.29 | 84.50 |
| | 2 | 61.94 | 91.11 | 96.00 | 41.00 | 75.29 | 56.55 | 68.50 |
| | 3 | 55.76 | 77.14 | 92.00 | 27.00 | 69.43 | 40.00 | 59.50 |
| | 4 | 53.89 | 69.70 | 90.00 | 23.00 | 67.42 | 34.59 | 56.50 |
| | 5 | 55.63 | 72.50 | 89.00 | 29.00 | 68.46 | 41.43 | 59.00 |
| | 6 | 50.29 | 51.72 | 86.00 | 15.00 | 63.47 | 23.26 | 50.50 |
| | 7 | 52.10 | 60.61 | 87.00 | 20.00 | 65.17 | 30.08 | 53.50 |
| | 8 | 54.07 | 75.00 | 93.00 | 21.00 | 68.38 | 32.81 | 57.00 |
| | 9 | 49.12 | 44.83 | 84.00 | 13.00 | 61.99 | 20.16 | 48.50 |
| | 10 | 50.00 | 50.00 | 83.00 | 17.00 | 62.41 | 25.37 | 50.00 |
| | 11 | 47.34 | 35.48 | 80.00 | 11.00 | 59.48 | 16.79 | 45.50 |
| | **Average** | **54.31** | **79.53** | **88.82** | 36.77 | **67.40** | 50.29 | **60.62** |

