# OpenReview forum: "Causal Reasoning Favors Encoders: Limits of Decoder-Only Models"
_ICLR.cc/2026/Conference — Submitted to ICLR 2026_

### Official Review · Reviewer_ivZq · 2025-10-18

**Soundness:** 3
**Presentation:** 2
**Contribution:** 3
**Rating:** 6
**Confidence:** 5

**Summary:**

This paper investigates the role of in-context learning (ICL) in LLMs' causal reasoning. Authors make the assumption that encoder and encoder-decoder architectures are better fitted for causal reasoning than the decoder-only architecture. To validate this assumption, the authors conduct experiments on the fine-tuned models with diverse ICL prompt strategies. Authors conclude that ICL alone is not sufficient for reliable causal reasoning, and encoder-only models are more suitable for causal reasoning. The current version has some minor writing problems.

**Strengths:**

1. I think the studied problem is novel and meaningful. Because both ICL and causal reasoning are important aspects of LLMs, it is interesting to link them. In addition, the influence of architectures on the causal reasoning abilities is worthy of exploring.

2. The authors drew an interesting conclusion: the encoder-only architecture has more generalized abilities. I think this conclusion has the potential to guide the design of advanced reasoning models.

3. The experiments are comprehensive, authors compare with diverse models. In addition, the experimental method is overall sound.

**Weaknesses:**

1. The first apparent weakness, I think, lies in the writing. For example, the first two paragraphs of the introduction are somewhat dispersed. I think authors can merge them into one paragraph, with the importance of causal reasoning coming first, then the ICL's influence on causal reasoning.

2. Second, I believe the transition between the first two paragraphs and the third paragraph is abrupt. Specifically, how did the ICL's influence on the causal reasoning connect to the comparison between the decoder-only and encoder architectures?

3. There should be a comma after the "To test this" in row 081.

4. Since the main results (conclusion) are related to the architectures rather than ICL, I suggest authors revise their introduction into an architecture-centric style rather than the current ICL-centric style.

5.  I believe Figure 1 should be revised. Specifically, authors can demonstrate a more detailed creation of their dataset in (A). In addition, I didn't get the main idea of the authors' evaluation methods in (B).

6. There should be different apparent sections in the related works section. For example, the LLM's causal reasoning part, the architecture part.

**Questions:**

Please refer to the weakness part.

---

> ### Author Response · Authors · 2025-11-29
>
> We thank the reviewer for their comments.
>
> >The first apparent weakness, I think, lies in the writing. For example, the first two paragraphs of the introduction are somewhat dispersed. I think authors can merge them into one paragraph, with the importance of causal reasoning coming first, then the ICL's influence on causal reasoning.
>
> >Second, I believe the transition between the first two paragraphs and the third paragraph is abrupt. Specifically, how did the ICL's influence on the causal reasoning connect to the comparison between the decoder-only and encoder architectures?
>
> >There should be a comma after the "To test this" in row 081.
>
> >Since the main results (conclusion) are related to the architectures rather than ICL, I suggest authors revise their introduction into an architecture-centric style rather than the current ICL-centric style.
>
> > There should be different apparent sections in the related works section. For example, the LLM's causal reasoning part, the architecture part.
>
> Thank you for the suggestions. We agree that it is important to highlight the significance of the choice of architecture in the paper. We will incorporate these suggestions for the introduction, related works in the updated version of the paper.

---

### Official Review · Reviewer_vzTB · 2025-10-28

**Soundness:** 3
**Presentation:** 3
**Contribution:** 2
**Rating:** 4
**Confidence:** 2

**Summary:**

This paper investigates whether encoder-based architectures  outperform decoder-only LLMs  in causal reasoning.  Using a synthetic first-order-logic dataset (SimpleLogic), the authors compare zero/few-shot ICL and fine-tuned settings under both natural-language (NL) and randomized (NNL) variants.  Results show that encoder and encoder–decoder models generalize more robustly than small-to-medium decoder-only models, though GPT-5 reaches near-perfect accuracy at much higher computational cost.

**Strengths:**

1. Clear and reproducible experimental setup with OOD splits.
2. Systematic architectural comparison (encoder / decoder / hybrid) under unified prompts.
3. The NNL (“lexical ablation”) split provides a neat way to isolate structural reasoning from lexical bias.

**Weaknesses:**

1. “Causal reasoning” is operationalized as deterministic logic inference; no interventional or counterfactual dimension.
2. Direct label prediction is insufficient to measure reasoning; CoT or reasoning-path supervision is absent.
3. Findings largely reproduce known trends from mathematical reasoning benchmarks.

**Questions:**

1. Would chain-of-thought distillation or reasoning supervision close the encoder–decoder gap?
2. How sensitive are results to dataset tokenization or prompt template?

---

> ### Author Response · Authors · 2025-11-29
>
> We thank the reviewer for their comments.
> >“Causal reasoning” is operationalized as deterministic logic inference; no interventional or counterfactual dimension.
>
> Our intention was to use logical reasoning as a surrogate task for certain aspects of causal reasoning, specifically the structured, rule-based inference component that appears in many causal systems. We agree that logical deduction does not, on its own, capture the full breadth of causal reasoning (particularly interventional or counterfactual reasoning) and we agree that these are fundamental distinctions in the broader causal inference literature.
>
> We will revise the framing to be more precise and avoid overstating the causal interpretation. We agree that it is more accurate and academically rigorous to focus on investigating logical reasoning; all while noting that these logical skills are foundational to certain forms of causal reasoning. Additionally, we will update the introduction and discussion to make this distinction explicit and clearly state that our operationalization focuses on logical inference rather than full causal reasoning.
>
>
> >Direct label prediction is insufficient to measure reasoning; CoT or reasoning-path supervision is absent.
>
> Thank you for the comment. We would like to clarify that this concern does not apply to our setup. Our fine-tuning explicitly includes both components: the reasoning path (proof chain) and the final prediction. The models are therefore supervised not only on the direct label but also on the intermediate reasoning steps, ensuring that the evaluation reflects actual reasoning ability rather than surface-level label prediction. We will make this clearer in the paper to avoid any ambiguity.
>
> > Question 1. Would chain-of-thought distillation or reasoning supervision close the encoder–decoder gap?
>
> Thank you for the question. In our experiments, all fine-tuned decoder-only baselines already receive reasoning-path supervision (proof chains + final answer), so our setup does incorporate a form of chain-of-thought supervision. Additionally, we evaluated large non-fine-tuned reasoning-oriented models (e.g., GPT-5-class models) that already internalize strong reasoning capabilities. These models outperform our fine-tuned models, but they do so with significantly higher inference cost and computational resources. This suggests that while chain-of-thought abilities can improve decoder-only performance, they do not eliminate the efficiency gap highlighted in our work.

---

> > ### Author Response · Authors · 2025-11-29
> >
> > > Question 2. How sensitive are results to dataset tokenization or prompt template?
> >
> > Tokenization and prompting do have some effect on results. Note that each fine-tuned model uses its own native tokenizer, and we indeed observe that model accuracy can drop on the NNL split, suggesting some sensitivity to tokenization differences. To further assess sensitivity to prompt formatting, we conducted automatic prompt optimization experiments using GPT-4.1 on both dataset splits (results shown in the table below)-
> >
> > Results for NL Dataset
> > | depth | precision_0 | precision_1 | recall_0 | recall_1 | f1_0 | f1_1 | accuracy |
> > |-------|-------------|-------------|----------|----------|------|------|----------|
> > | 0     | 0.000000    | 1.000000    | 0.000000 | 1.000000 | 0.000000 | 1.000000 | 1.000000 |
> > | 1     | 0.944785    | 1.000000    | 1.000000 | 0.938356 | 0.971609 | 0.968198 | 0.970000 |
> > | 2     | 0.733010    | 0.925532    | 0.955696 | 0.612676 | 0.829670 | 0.737288 | 0.793333 |
> > | 3     | 0.613169    | 0.929825    | 0.973856 | 0.360544 | 0.752525 | 0.519608 | 0.673333 |
> > | 4     | 0.525692    | 0.851064    | 0.950000 | 0.250000 | 0.676845 | 0.386473 | 0.576667 |
> > | 5     | 0.573705    | 0.857143    | 0.953642 | 0.281879 | 0.716418 | 0.424242 | 0.620000 |
> > | 6     | 0.540856    | 0.860465    | 0.958621 | 0.238710 | 0.691542 | 0.373737 | 0.586667 |
> > | 7     | 0.537255    | 0.666667    | 0.901316 | 0.202703 | 0.673219 | 0.310881 | 0.556667 |
> > | 8     | 0.568627    | 0.600000    | 0.889571 | 0.197080 | 0.693780 | 0.296703 | 0.573333 |
> > | 9     | 0.548117    | 0.721311    | 0.885135 | 0.289474 | 0.677003 | 0.413146 | 0.583333 |
> > | 10    | 0.529412    | 0.444444    | 0.843750 | 0.142857 | 0.650602 | 0.216216 | 0.516667 |
> > | 11    | 0.488000    | 0.340000    | 0.787097 | 0.117241 | 0.602469 | 0.174359 | 0.463333 |
> > | **Average** | **0.586220** | **0.857143** | **0.917213** | **0.434149** | **0.715281** | **0.576365** | **0.659444** |
> >
> > Results for NNL Dataset
> > | depth | precision_0 | precision_1 | recall_0 | recall_1 | f1_0 | f1_1 | accuracy |
> > |-------|-------------|-------------|----------|----------|------|------|----------|
> > | 0     | 0.000000 | 1.000000 | 0.000000 | 1.000000 | 0.000000 | 1.000000 | 1.000 |
> > | 1     | 0.930693 | 0.939394 | 0.940000 | 0.930000 | 0.935323 | 0.934673 | 0.935 |
> > | 2     | 0.705357 | 0.761364 | 0.790000 | 0.670000 | 0.745283 | 0.712766 | 0.730 |
> > | 3     | 0.586538 | 0.593750 | 0.610000 | 0.570000 | 0.598039 | 0.581633 | 0.590 |
> > | 4     | 0.623853 | 0.648352 | 0.680000 | 0.590000 | 0.650718 | 0.617801 | 0.635 |
> > | 5     | 0.590476 | 0.600000 | 0.620000 | 0.570000 | 0.604878 | 0.584615 | 0.595 |
> > | 6     | 0.585586 | 0.606742 | 0.650000 | 0.540000 | 0.616114 | 0.571429 | 0.595 |
> > | 7     | 0.586207 | 0.566372 | 0.510000 | 0.640000 | 0.545455 | 0.600939 | 0.575 |
> > | 8     | 0.573034 | 0.558559 | 0.510000 | 0.620000 | 0.539683 | 0.587678 | 0.565 |
> > | 9     | 0.573333 | 0.544000 | 0.430000 | 0.680000 | 0.491429 | 0.604444 | 0.555 |
> > | 10    | 0.514706 | 0.507576 | 0.350000 | 0.670000 | 0.416667 | 0.577586 | 0.510 |
> > | 11    | 0.602740 | 0.559055 | 0.440000 | 0.710000 | 0.508671 | 0.625551 | 0.575 |
> > | **Average** | **0.631528** | **0.672767** | **0.593636** | **0.706923** | **0.611996** | **0.689422** | **0.655** |
> >
> >
> > We observe minimal improvements from prompt optimization, suggesting that prompt-template sensitivity does not significantly affect the overall comparative conclusions of the paper.

---

### Official Review · Reviewer_157V · 2025-11-01

**Soundness:** 3
**Presentation:** 3
**Contribution:** 2
**Rating:** 6
**Confidence:** 3

**Summary:**

In this paper, the authors experimentally demonstrate that decoder-only language models exhibit weaker logical reasoning abilities compared to fine-tuned encoder and encoder-decoder models. They intuitively attribute this observation to the fact that encoder layers allow every token to integrate information from the entire sequence, thereby enhancing the model's multi-hop conjunctive reasoning capabilities.

**Strengths:**

1. Investigating the logical reasoning abilities of LLMs is an important research direction that can contribute to enhancing the interpretability and trustworthiness of LLMs.

2. The paper is clearly written and easy to follow.

3. The authors have made the code and data publicly available, which greatly facilitates reproducibility and further research.

4. The authors’ finding that decoder-only language models exhibit weaker logical reasoning abilities compared to fine-tuned encoder and encoder-decoder models is both intriguing and intuitively reasonable. Their understanding that models with an encoder possess stronger multi-hop conjunctive reasoning capabilities adds valuable insights.

**Weaknesses:**

1. In my opinion, the paper’s focus may not be entirely well-positioned, as the problem definition, dataset construction, and experimental validation all primarily center around logical reasoning, rather than causal reasoning. Given the fundamental differences between causal reasoning and logical reasoning [1], I believe it would be more precise and academically rigorous to reframe the study from causal reasoning to logical reasoning.

2. The paper proposes an interesting finding: decoder-only language models exhibit weaker logical reasoning abilities compared to fine-tuned encoder and encoder-decoder models, which I appreciate. However, as the authors mention, very large decoder-only models such as GPT-5 still demonstrate significant out-of-distribution (OOD) generalization abilities, despite their lower efficiency. A more impactful and forward-looking contribution would be to explore how to improve encoder models, or integrate concepts from encoder models into decoder-only models, aiming to reduce the complexity of LLMs while enhancing their logical reasoning capabilities.

[1] Hernán, M. A., & Robins, J. M. (2010). Causal inference.

**Questions:**

Please see Weaknesses.

---

> ### Author Response · Authors · 2025-11-29
>
> We thank the reviewer for their comments.
> > In my opinion, the paper’s focus may not be entirely well-positioned, as the problem definition, dataset construction, and experimental validation all primarily center around logical reasoning, rather than causal reasoning. Given the fundamental differences between causal reasoning and logical reasoning [1], I believe it would be more precise and academically rigorous to reframe the study from causal reasoning to logical reasoning.
>
> Our intention was to use logical reasoning as a surrogate task for certain aspects of causal reasoning, specifically the structured, rule-based inference component that appears in many causal systems. We agree that logical deduction does not, on its own, capture the full breadth of causal reasoning (particularly interventional or counterfactual reasoning) and we agree that these are fundamental distinctions in the broader causal inference literature.
>
> We will revise the framing to be more precise and avoid overstating the causal interpretation. We agree that it is more accurate and academically rigorous to focus on investigating logical reasoning; all while noting that these logical skills are foundational to certain forms of causal reasoning. Additionally, we will update the introduction and discussion to make this distinction explicit and clearly state that our operationalization focuses on logical inference rather than full causal reasoning.
>
>
> > The paper proposes an interesting finding: decoder-only language models exhibit weaker logical reasoning abilities compared to fine-tuned encoder and encoder-decoder models, which I appreciate. However, as the authors mention, very large decoder-only models such as GPT-5 still demonstrate significant out-of-distribution (OOD) generalization abilities, despite their lower efficiency. A more impactful and forward-looking contribution would be to explore how to improve encoder models, or integrate concepts from encoder models into decoder-only models, aiming to reduce the complexity of LLMs while enhancing their logical reasoning capabilities.
>
> We agree that this is a good direction to follow. However, it is beyond the scope of the current paper for two reasons. First, doing so would require substantial methodological work (e.g., new training protocols, hybrid architectures, or large-scale distillation/retrofit experiments) that would meaningfully expand the paper’s focus. Second, these experiments demand large-scale compute and engineering resources (training from scratch or extensive fine-tuning/distillation across many model sizes and datasets), which exceed our current computational budget.

---

### Official Review · Reviewer_CfHs · 2025-11-04

**Soundness:** 3
**Presentation:** 2
**Contribution:** 2
**Rating:** 4
**Confidence:** 3

**Summary:**

This paper investigates the limitations of decoder-only large language models (LLMs) in causal reasoning tasks, arguing that encoder and encoder-decoder architectures are better suited for multi-hop conjunctive reasoning due to their ability to project inputs into a latent space and perform global information aggregation. The authors introduce a benchmark based on SimpleLogic, with natural-language (NL) and non-natural-language (NNL) test sets, and evaluate a range of models under zero-shot, few-shot, and fine-tuned settings. They find that fine-tuned encoder-based models outperform decoder-only models in efficiency and robustness, especially under distributional shift, while very large decoders (e.g., GPT-5) achieve high accuracy at significant computational cost.

**Strengths:**

1. The paper provides a clear and motivated comparison of architectural families for causal reasoning, with a well-designed dataset that controls for linguistic and structural variability.

2. The inclusion of both natural and non-natural language splits strengthens the evaluation of generalization and robustness.

3. The analysis is thorough, covering accuracy, depth-wise performance, label compliance, and computational cost.

**Weaknesses:**

1. The abstract could be more concise. For example, the sentence “We hypothesize that, due to their ability to project the input into a latent space, encoder- and encoder-decoder architectures are better suited for said multi-hop conjunctive reasoning versus decoder-only models” is somewhat verbose and could be streamlined.

2. The theoretical argument in Section 3.2, while intuitive, lacks formal rigor and could benefit from a more structured comparison of the representational capacities of encoder vs. decoder architectures.

3. The operationalization of “causal reasoning” is largely limited to logical deduction in FOL. The authors should discuss whether this adequately captures real-world causal reasoning and how their findings generalize to broader causal settings (e.g., intervention, counterfactuals).

4. The naming of the dataset (“NL Depth-12”, “NNL Depth-12”) is functional but not memorable. A more distinctive name (e.g., “LogicCausal-Bench”) could improve recognition and citation.

5. While the paper mentions related benchmarks (e.g., SimpleLogic), it does not explicitly position itself against recent causal reasoning benchmarks (e.g., CLADDER, CLEAR). A dedicated comparison would help clarify the novelty and scope of the proposed evaluation.

6. Several figures (e.g., Figure 3, 4, 5) use small text and light colors, reducing readability. Simplifying the visualizations and using higher contrast would improve clarity.

**Questions:**

1. How does the performance of fine-tuned encoder models compare with very large decoder models when controlling for computational budget?

2. Could the advantage of encoders be attributed to pretraining data/style rather than architecture alone?

3. Would the results hold in more realistic causal settings involving interventions or counterfactuals?

---

> ### Author Response · Authors · 2025-11-29
>
> We thank the reviewer for the comments.
>
> >The abstract could be more concise. For example, the sentence “We hypothesize that, due to their ability to project the input into a latent space, encoder- and encoder-decoder architectures are better suited for said multi-hop conjunctive reasoning versus decoder-only models” is somewhat verbose and could be streamlined.
>
> >While the paper mentions related benchmarks (e.g., SimpleLogic), it does not explicitly position itself against recent causal reasoning benchmarks (e.g., CLADDER, CLEAR). A dedicated comparison would help clarify the novelty and scope of the proposed evaluation.
>
> >Several figures (e.g., Figure 3, 4, 5) use small text and light colors, reducing readability. Simplifying the visualizations and using higher contrast would improve clarity.
>
> We thank the reviewer for these suggestions, we will incorporate these changes in an updated version of the paper.
>
> ---
>
> >The theoretical argument in Section 3.2, while intuitive, lacks formal rigor and could benefit from a more structured comparison of the representational capacities of encoder vs. decoder architectures.
>
> Broader theoretical perspective is out of the scope for this paper. However, to address this particular weakness , we have added a new section - Section 7: MECHANISTIC INTERPRETABILITY OF LOGICAL FLOW, which clearly shows the representational capacities of encoder vs. decoder architectures.
>
> >The operationalization of “causal reasoning” is largely limited to logical deduction in FOL. The authors should discuss whether this adequately captures real-world causal reasoning and how their findings generalize to broader causal settings (e.g., intervention, counterfactuals).
>
> Our intention was to use logical reasoning as a surrogate task for certain aspects of causal reasoning, specifically the structured, rule-based inference component that appears in many causal systems. We agree that logical deduction does not, on its own, capture the full breadth of causal reasoning (particularly interventional or counterfactual reasoning) and we agree that these are fundamental distinctions in the broader causal inference literature.
>
> We will revise the framing to be more precise and avoid overstating the causal interpretation. We agree that it is more accurate and academically rigorous to focus on investigating logical reasoning; all while noting that these logical skills are foundational to certain forms of causal reasoning. Additionally, we will update the introduction and discussion to make this distinction explicit and clearly state that our operationalization focuses on logical inference rather than full causal reasoning.
>
>
> > Question 1. How does the performance of fine-tuned encoder models compare with very large decoder models when controlling for computational budget?
>
> Answer: We thank the reviewer for pointing this out. In terms of performance versus inference cost, we found that fine-tuned encoder models outperform very large decoder models (Table shown below). However, in terms of FLOPS/latency on hardware/etc, such a comparison it is not possible, particularly due to the opaque nature of the larger models. We will add a brief discussion on this on the updated version of the paper.
>
> | Model          | Inference Time (hours; ↓) | Efficiency (Accuracy/Hour; ↑) | Hardware        |
> |----------------|--------------------------|--------------------------------|-----------------|
> | BART-Base      | 0.1                      | 640                            | Nvidia RTX 6000 |
> | Flan-T5-Base   | 0.45                     | 143.4                          | Nvidia RTX 6000 |
> | BERT-Base      | 0.17                     | 388.2                          | Nvidia RTX 6000 |
> | Qwen3-1.7B     | 4.9                      | 12.9                           | Nvidia RTX 6000 |
> | GPT-5          | 90.6                     | 1.1                            | API             |
> | GPT-4.1        | 0.5                      | 129                            | API             |
> | Claude Opus 4.1| 14.4                     | 5.5                            | API             |
>
>
> > Question 2. Could the advantage of encoders be attributed to pretraining data/style rather than architecture alone?
>
> Answer: We agree that it could be, but with a nuance. Fully ablating pretraining data and strategy is challenging, as existing encoder and decoder models are not trained on matched datasets or with identical objectives. In fact, we could hypothesize that the datasets used for the older models is present in the newer decoder-only models (albeit, naturally, this is only a hypothesis and pretraining objectives also matter). While the encoder advantage we observe could partially stem from differences in pretraining data/style, we have, however, mitigated this by finetuning the models on the same datasets / task objectives.

---

### Author Response · Authors · 2025-12-03
**Dear Reviewers (rebuttal)**

We thank the reviewers for the feedback. We have uploaded an updated version of the paper incorporating all the changes suggested by the reviewers.
**In this post:**
- (1) We address a common concern raised by reviewers CfHs, 157V and vzTB.
- (2) We summarize the changes to the updated PDF document.

## (1) Concern raised by reviewers CfHs, 157V and vzTB.
`CfHs`: *"The operationalization of “causal reasoning” is largely limited to logical deduction in FOL. The authors should discuss whether this adequately captures real-world causal reasoning and how their findings generalize to broader causal settings (e.g., intervention, counterfactuals)."*

`157V`: *"In my opinion, the paper’s focus may not be entirely well-positioned, as the problem definition, dataset construction, and experimental validation all primarily center around logical reasoning, rather than causal reasoning. Given the fundamental differences between causal reasoning and logical reasoning [1], I believe it would be more precise and academically rigorous to reframe the study from causal reasoning to logical reasoning."*

`vzTB`: *"“Causal reasoning” is operationalized as deterministic logic inference; no interventional or counterfactual dimension."*

Our intention was to use logical reasoning as a surrogate task for certain aspects of causal reasoning, specifically the structured, rule-based inference component that appears in many causal systems. We agree that logical deduction does not, on its own, capture the full breadth of causal reasoning (particularly interventional or counterfactual reasoning) and we agree that these are fundamental distinctions in the broader causal inference literature.
We have revised the framing (in the updated version) to be more precise and avoid overstating the causal interpretation. We agree that it is more accurate and academically rigorous to focus on investigating logical reasoning; all while noting that these logical skills are foundational to certain forms of causal reasoning. Additionally, we will update the introduction and discussion to make this distinction explicit and clearly state that our operationalization focuses on logical inference rather than full causal reasoning.

## (2) Changes to the PDF
### Main Paper
| Category   | Added Content | Removed Content |
|-----------|---------------|---------------|
| **Sections** | 6. Ablations (Addressing Reviewer `CfHs`'s Question 1 ) |
|           | 7. Mechanistic Interpretability of Logical Flow ( Addressing Reviewer `CfHs`'s 2nd weakness )|
| **Paragraphs** | Section 1. Introduction: The first two paragraphs have been changed as per Reviewer `ivZq`'s suggestion and reflect our studies emphasis on logical reasoning (common concern for reviewers `CfHs`, `157V` and `vzTB` ) |
|           | Section 2. Related Works: apparent sections (as suggested by Reviewer `ivZq` ) and more emphasis on Logical reasoning in LLMs and impact of architectures (common concern for reviewers `CfHs`, `157V` and `vzTB` ) |
|           |Names of Section 3.1 and 3.2 have been changed to reflect our emphasis on logical reasoning (common concern for reviewers `CfHs`, `157V` and `vzTB` ) |
| **Figures** | Figure 3: Combined label response frequency bar plots (values in percentage (%)) (Reviewer `CfHs` concerns regarding readability)| Figure 3: Comparison of average accuracy all non-finetuned models|
|           | Figure 4: Depth-wise accuracy plots (added with results for Flan-T5 Base) | Figure 4 and 5 (Comparison of label prediction frequencies of models)|
|           | Figure 5: Mechanistic Analysis |Figure 7:  Side-by-side comparison of average accuracy across Reasoning models. |
| **Tables** | Table 1: Inference times and efficiency |  Table 1 and 2 (Table 2: AUC values of models) |
|                           |                              | Table 3: Average inference time (hr) |

### Appendix
| Category   | Added Content | Removed Content |
|-----------|---------------|---------------|
| **Sections** | Section B.6 Learning Curves (of the finetuned models) |   |
|           | Section B.7 Evaluating Prompt Sensitivity (Addressing Reviewer `vzTB`'s Question 2) | |
|           | Section B.3 Finetuning: We have explained our finetuning methodology in detail (addressing Reviewer `vzTB`'s concern- *reasoning-path supervision is absent.*)
|           |Section C (AU/ROC analysis) has been rewritten |
| **Figures** | Figure 7, 8 Combined ROC curves | Figure 8,9,10,11 (Individual ROC curves) |
|           | Figure 9: Depth-wise accuracy plots for decoder-only models | Figure 4 and 5 (Comparison of label prediction frequencies of non-finetuned and finetuned models)|
|           | Figure 10 and 12: Depth-wise metric plots (precision, recall, F1) for Decoder-only models | |
|           | Figure 11 and 13: Depth-wise metric plots (precision, recall, F1) for finetuned models | |
| **Tables** | Table 5: Automatic Prompt Optimization Results (Addressing Reviewer `vzTB`'s Question 2) |   |

---

### Meta-Review · Area_Chair_GJ3S · 2026-01-07

**Summary:**

This paper investigates the limitations of decoder-only large language models (LLMs) in causal reasoning tasks, arguing that encoder and encoder-decoder architectures are better suited for multi-hop conjunctive reasoning due to their ability to project inputs into a latent space and perform global information aggregation.

The main concerns from the reviewers include:

1) Unclear position of this paper. The operationalization of “causal reasoning” is largely limited to logical deduction in FOL. The authors should discuss whether this adequately captures real-world causal reasoning and how their findings generalize to broader causal settings (e.g., intervention, counterfactuals).

2) Unclear presentation across the whole paper.

3) Insight for future action is not clear and well-discussed. Very large decoder-only models such as GPT-5 still demonstrate significant out-of-distribution (OOD) generalization abilities, despite their lower efficiency. A more impactful and forward-looking contribution would be to explore how to improve encoder models, or integrate concepts from encoder models into decoder-only models, aiming to reduce the complexity of LLMs while enhancing their logical reasoning capabilities.

4) CoT or reasoning-path supervision is absent. Direct label prediction is insufficient to measure reasoning.

**Reviewer Concerns:**

The above concerns are important to further confirm the contributions of this paper. However, unfortunately, these concerns remain even after the rebuttal.

For the first one, the authors admitted this drawback, but the solution might not be ideal. From all reviews, it might be better to reform this paper by following logical reasoning, instead of causal reasoning where the most important parts are intervention, counterfactuals but not considered in this paper. The second concern remains if the first one is not addressed.

For the third one, I think it is important to have discussion here, but the authors thought it is out of scope.

For the fourth one, the authors might misunderstand reviewer's meaning. Although the label prediction in this paper can reflect the reasoning quality, it is just one of the reasoning cases. Considering evaluation on the reasoning path is also promising and could be done later.

**Reviewer Scores:**

The main concerns still remain, thus the score is at most 4,4,6,6, which is a borderline paper. However, based on the above meta review, this paper still needs a major revision to find a better position to demonstrate its contributions. Thus, I recommend rejection on this round.

---

### Decision · Program_Chairs · 2026-01-26

Reject